# ELISA detection of MPO-DNA complexes in human plasma is error-prone and yields limited information on neutrophil extracellular traps formed *in vivo*

Hubert Hayden[1], Nahla Ibrahim[1], Johannes Klopf[1], Branislav Zagrapan[1], Lisa-Marie Mauracher[2], Lena Hell[2], Thomas M. Hofbauer[3], Anna S. Ondracek[3], Christian Schoergenhofer[4], Bernd Jilma[4], Irene M. Lang[3], Ingrid Pabinger[2], Wolf Eilenberg[1], Christoph Neumayer[1], Christine Brostjan[1]*

1 Division of Vascular Surgery, Department of General Surgery, Medical University of Vienna, Vienna General Hospital, Vienna, Austria, 2 Division of Hematology and Hemostaseology, Department of Internal Medicine I, Medical University of Vienna, Vienna General Hospital, Vienna, Austria, 3 Division of Cardiology, Department of Internal Medicine II, Medical University of Vienna, Vienna General Hospital, Vienna, Austria, 4 Department of Clinical Pharmacology, Medical University of Vienna, Vienna General Hospital, Vienna, Austria

* christine.brostjan@meduniwien.ac.at

**Data Availability Statement:** All relevant data are within the manuscript and its Supporting Information files.

## Abstract

Over the past years, neutrophil extracellular traps (NETs) were shown to contribute to states of acute and chronic inflammatory disease. They are composed of expelled chromatin and decorated by neutrophil-derived proteins. Therefore, the analysis of DNA complexes with myeloperoxidase (MPO) by ELISA has become an attractive tool to measure NET formation in *in vitro* and *in vivo* samples. When we used a published MPO-DNA ELISA protocol and included an isotype control for the anti-MPO coating antibody, we observed high assay specificity for *in vitro* prepared NET samples, whereas the specificity for *in vivo* plasma samples was low. In addition, the assay failed to detect *in vitro* generated MPO-DNA complexes when spiked into plasma. Therefore, we set out to improve the specificity of the MPO-DNA ELISA for plasma samples. We found that the use of Fab fragments or immunoglobulins from different species or reversal of the antibody pair led to either a high background or a low dynamic range of detection that did not improve the specificity for plasma samples. Also, the use of higher plasma dilutions or pre-clearing of plasma immunoglobulins were ineffective. Finally, we found that a commercial reagent designed to block human anti-mouse antibodies and multivalent substances increased the detection window between the MPO antibody and isotype control for highly diluted plasma. We applied this modified ELISA protocol to analyze MPO-DNA complexes in human blood samples of acute and chronic inflammatory conditions. While markers of neutrophil activation and NET formation such as MPO, elastase and citrullinated histone H3 correlated significantly, we observed no correlation with the levels of MPO-DNA complexes. Therefore, we conclude that ELISA measurements of MPO-DNA complexes in human plasma are highly questionable regarding specificity of NET detection. In general, plasma analyses by ELISA should more frequently include isotype controls for antibodies to demonstrate target specificity.

**Funding:** This work was primarily supported by the Austrian Science Fund (SFB subproject F 5409-B21 issued to CB). The funders had no role in study design, data collection and analysis, decision to publish, or preparation of the manuscript.

**Competing interests:** The authors have declared that no competing interests exist.

## Introduction

Neutrophil extracellular trap (NET) formation is a type of cell death distinct from apoptosis and necrosis [1, 2] that extends the classical neutrophil defense strategies of phagocytosis and degranulation [3]. NETs have been originally characterized as web-like extracellular structures composed of nuclear DNA, histones and neutrophil granule proteins resulting in a high local concentration of antimicrobial substances to entrap and destroy pathogens [4].

Reactive oxygen species (ROS) have emerged as key players in NET formation [5]. Nicotinamide adenine dinucleotide phosphate (NADPH) oxidase 2 (NOX2)-triggered ROS stimulate the nuclear translocation of two neutrophil granule proteins, neutrophil elastase (NE) and myeloperoxidase (MPO). NE and MPO are major mediators of histone degradation and chromatin decondensation and thus promote NET formation [6, 7]. Decondensation of nuclear DNA and subsequent NET release can as well be induced by peptidylarginine deiminase 4 (PADI4)-driven histone citrullination [8].

While extracellular deposition of DNA, histones and neutrophil proteases serves a beneficial purpose in microbial defense, it may entail collateral tissue damage [9]. Detrimental effects of NETs have been reported for septic conditions [10] as well as sterile pathologies such as autoimmune diseases [11, 12], thrombosis [13, 14] and cancer [15, 16]. Of note, NETs were shown to contribute to the development of abdominal aortic aneurysms (AAAs) [17–19].

Thus, NET analysis of patient specimens has been pursued to characterize the role of NETs in disease and to possibly delineate biomarkers and therapeutic approaches. Since access to tissue samples is limited, blood and plasma parameters of NET formation serve as attractive markers. However, care must be taken regarding the choice of appropriate plasma molecules reflecting *in vivo* NET release. Analysis of circulating cell-free DNA [14, 17] or nucleosomes [20–22] is not considered specific for NET formation, as they may also originate from necrotic processes. Assessment of circulating neutrophil granule proteins faces similar limitations, since their release into plasma is not restricted to NET expulsion but generally occurs during neutrophil degranulation. Citrullinated histones are a strong evidence of NET formation, but represent only one pathway of NET induction, while other PADI4-independent mechanisms of NET formation do exist [2, 23]. Therefore, the detection of extracellular DNA in complex with neutrophil proteins is promising, because these complexes are mainly expelled during NET formation and are less likely to occur incidentally by molecule interactions in plasma. Among these, complexes of MPO [11, 12, 17, 20, 24] or NE [12, 25, 26] with DNA have frequently been studied in clinical samples. Most protocols for assessment of neutrophil protein-DNA complexes share a common sandwich enzyme-linked immunosorbent assay (ELISA) concept. Both, uncomplexed and DNA-bound target proteins are first captured onto an antibody-coated solid surface. Subsequently, the neutrophil protein-DNA complexes are detected using a labeled antibody raised against DNA. While the detection of neutrophil protein-DNA complexes is considered to have a high specificity for NET formation and to offer the advantage of a simple quantification [27], no consensus for standardized assessment of NET formation exists.

Thus, with a particular interest in clinical application, we aimed to adopt and evaluate published ELISA protocols for blood-borne parameters of NET formation. On the basis of frequent reporting, MPO-DNA complexes were selected for detailed evaluation. Coating and detection antibodies, sample dilution as well as basic ELISA procedures were adopted from the literature [12, 28]. When we replaced the specific anti-MPO coating antibody with a corresponding isotype control, we observed high ELISA specificity and sensitivity for *in vitro* generated NET samples, while the MPO-DNA specificity for *in vivo* retrieved plasma samples was low. Since comparable controls for assay specificity were not included in previously published

studies, the respective MPO-DNA ELISA protocols have been applied to measure NETs in plasma or serum despite this severe assay limitation. By introducing various protocol adaptions, we aimed to increase the specificity of the assay for MPO-DNA complexes in plasma samples and to apply the modified ELISA protocol for NET assessment in pathological conditions in comparison to other markers of neutrophil activation and NET formation.

## Materials and methods

### Calibrator preparation

The supernatant of neutrophils exposed to phorbol 12-myristate 13-acetate (PMA) or solvent control served as calibrator for the MPO-DNA complex ELISA. Neutrophils were isolated from ethylenediaminetetraacetic acid (EDTA)-anticoagulated peripheral blood by means of density gradient centrifugation with modifications to a previously published protocol [29]. Briefly, 3 ml of Ficoll®-Paque Plus (GE Healthcare, Chicago, IL) were layered onto 3 ml of Histopaque®-1119 (Sigma-Aldrich, St. Louis, MO) in a 15 ml polypropylene centrifugation tube. 9 ml of blood were then added and centrifuged at 700 x g for 30 min at room temperature without brakes. The neutrophil layer was removed and washed once with phosphate-buffered saline (PBS) without calcium and magnesium (Thermo Fisher Scientific, Waltham, MA). For erythrocyte lysis, the cells were gently resuspended in 6.5 ml hypotonic PBS (1:5.2 diluted with distilled water) for 90 sec. Isotonicity was re-established by addition of 2.2 ml of a 3% sodium chloride solution. Neutrophils were counted with a Sysmex XN-350 device (Sysmex, Kobe, Japan) and adjusted to a concentration of $2 \times 10^6$ cells per ml with Hank's buffered saline solution (HBSS) with calcium and magnesium (Lonza, Basel, Switzerland). Cells were stimulated in 1.5 ml reaction tubes with 1 µg/ml (1.621 µM) PMA (Sigma-Aldrich) or dimethyl sulfoxide (DMSO, Sigma-Aldrich) as solvent control at 37˚C for 3 h. The supernatant was then collected and frozen in aliquots at -80˚C. Calibrator DNA content was assessed with Quant-iT™ Pico-Green™ dsDNA Assay Kit (Thermo Fisher Scientific). Calibrator MPO concentration was quantified with a commercially available assay (Human Myeloperoxidase Quantikine ELISA Kit, Bio-Techne, Minneapolis, MN) according to the manufacturer's recommendations.

### Initial (standard) MPO-DNA complex ELISA protocol

The basic ELISA procedure was adopted from the literature [12, 28]. ABTS ELISA Buffer Kit (Pepro Tech, Rocky Hill, NJ) served as a source of ancillary ELISA materials providing wash buffer, blocking buffer, sample diluent, 96-well microplates and sealing films. Microplates were coated overnight with 100 µl of diluted antibody per well at 4˚C on an orbital shaker (500 rpm). The following antibodies were diluted in PBS without calcium and magnesium to a concentration of 5 µg/ml for coating: mouse anti-human MPO monoclonal antibody (clone 4A4, IgG2b, verified for ELISA use, RRID:AB_617350, no. 0400–0002, Bio-Rad, Hercules, CA) or mouse IgG2b monoclonal isotype control (clone 20116, immunogen: KLH, RRID:AB_357346, no. MAB004, Bio-Techne). Where indicated, the antibody was omitted during the coating procedure. The next day, the antibody solution was removed and the plate was washed four times using 300 µl wash buffer per well and washing step. Wells were blocked with 300 µl of blocking buffer for 1 h at room temperature under agitation followed by repeated washing as indicated above. Calibrator and plasma samples (anticoagulated with citrate or the combination of citrate-theophylline-adenosine-dipyridamole [CTAD]) were diluted in sample diluent of the ABTS ELISA Buffer Kit. Plain sample diluent served as blank. 95 µl of calibrator or sample were applied onto the coated and blocked microplate per well and incubated for 2 h at room temperature on an orbital shaker (500 rpm). The DNA moiety was detected with a peroxidase-conjugated mouse anti-DNA monoclonal antibody (clone MCA-33, detecting single- and

double-stranded [ds] DNA, component number 2 of Cell Death Detection ELISA, no. 11544675001, Sigma-Aldrich). The lyophilized antibody was first dissolved according to the manufacturer's recommendations and then diluted 1:10.5 with incubation buffer (component number 5 of Cell Death Detection ELISA). After the microplate was again washed as indicated above, 100 µl of diluted detection antibody were applied per well and incubated for 1.5 h at room temperature under agitation. Next, excess detection antibody was removed with four wash cycles using wash buffer of the Cell Death Detection ELISA (component number 4, 300 µl per well). Signal was developed for 20 min on an orbital shaker (500 rpm) with 2,2'-azino-bis(3-ethylbenzothiazoline-6-sulfonic acid) (ABTS) substrate solution (composed of components number 6 and 7 of Cell Death Detection ELISA) and subsequently measured on a Varioskan Flash device (Thermo Fisher Scientific) set to 405 nm (main wavelength) and 490 nm (reference wavelength). Readings at 490 nm were subtracted from 405 nm readings. The calibrator dilution of 1:1600 was set to "1" in all assays to deduce relative sample concentrations. While calibrator dilutions mostly ranged from 1:50 to 1:1600, various plasma dilutions were tested throughout the study as indicated in the respective figure captions.

## Reagents used for ELISA optimization

The following reagents were used to modify the initial MPO-DNA complex ELISA protocol. Coating antibodies: rabbit anti-human MPO polyclonal antibody (immunogen: KLH-conjugated linear peptide corresponding to a sequence within human MPO heavy chain region, RRID: AB_310666, no. 07-496-I, EMD Millipore, Billerica, MA), mouse anti-ds DNA monoclonal antibody (clone 35I9, IgG2a kappa, validated for ELISA use, RRID:AB_470907, no. ab27156, Abcam, Cambridge, UK), mouse IgG2a kappa monoclonal isotype control (RRID: AB_840850, no. 010-001-332, Rockland Immunochemicals, Limerick, PA), mouse IgG2b monoclonal isotype control (clone eBMG2b, RRID:AB_470117, no. 14-4732-82, Thermo Fisher Scientific) and mouse IgG1 monoclonal isotype control (clone P3.6.2.8.1, RRID:AB_470110, no. 14-4714-81, Thermo Fisher Scientific). Alternative variant of wash buffer: washing buffer (component number 4 of Cell Death Detection ELISA). Alternative variants of blocking buffer: incubation buffer (component number 5 of Cell Death Detection ELISA), The Blocking Solution (Candor Bioscience, Wangen, Germany) as well as a self-made in-house blocking buffer composed of 0.1% bovine serum albumin (BSA, Sigma-Aldrich) and 5 mM EDTA (Sigma-Aldrich) dissolved in PBS without calcium and magnesium. Alternative variants of sample diluent: incubation buffer (component number 5 of Cell Death Detection ELISA), in-house blocking buffer, The Blocking Solution, LowCross-Buffer® Mild, LowCross-Buffer® and LowCross-Buffer® Strong (all Candor Bioscience). Additives for sample diluent: protein A-purified mouse IgG from normal mouse serum (no. A66185M, Meridian Life Science, Memphis, TN) and TRU Block™ Ready Heterophilic Antibody Interference Active Blocker (no. 8001, Meridian Life Science). Alternative detection antibodies and other reagents: horseradish peroxidase (HRP)-labeled mouse anti-human MPO monoclonal antibody (clone MPO421-8B2, IgG1, RRID:AB_2827763, no. NBP2-41406H, Bio-Techne), biotinylated mouse anti-human MPO monoclonal antibody (clone 266-6K1, IgG1, RRID:AB_10234434, no. HM2164BT, HyCult Biotech, Uden, The Netherlands) and Pierce™ High Sensitivity Streptavidin-HRP (Thermo Fisher Scientific). Detection antibody diluents: in-house blocking buffer, The Blocking Solution, LowCross-Buffer® Mild, LowCross-Buffer® and LowCross-Buffer® Strong.

## Fab fragment preparation

Coating antibodies of the initial MPO-DNA complex ELISA protocol were processed with Pierce™ Fab Micro Preparation Kit (Thermo Fisher Scientific) to Fab and Fc fragments

according to the manufacturer's recommendations. First, 125 µg of antibody in a volume of 125 µl were digested with 65 µl of agarose-immobilized papain in a spin-column (kit components) at 37˚C for 5 h under constant mixing. The sample was separated from the immobilized papain by double centrifugation at 5000 x g for 1 min. To further fraction the generated antibody fragments the solution was added to an equilibrated protein A column (kit component) and after 10 min of incubation, Fab fragments were retrieved by a centrifugation step at 1000 x g for 1 min (Fab fragment fraction 1). Furthermore, Fab fragment fractions 2 and 3 were obtained by washing the column twice with 200 µl of PBS and centrifuging again. Fc fragments bound to protein A (Fc fragment fractions 1, 2 and 3) were comparably collected by adding 400 µl of elution buffer to the column and centrifuging. 40 µl of neutralization buffer (kit component) were immediately added to each Fc fragment fraction. Protein content after purification was quantified using a Nano Drop 8000 spectrophotometer (Thermo Fisher Scientific).

## DNA digestion

Micrococcal nuclease (MNase) from *S. aureus* (Sigma-Aldrich) was added to undiluted calibrator or plasma samples at a final concentration of 2 or 20 U/ml and incubated for 60 min at 37˚C. Control samples were incubated at 37˚C or kept on ice without MNase supplementation. After digestion, calibrator and plasma samples were processed as indicated for the initial MPO-DNA complex ELISA protocol.

## Clearance of plasma immunoglobulins

Immunoglobulin (Ig) was removed from plasma according to a previously published protocol [30]. Protein A/G PLUS-agarose (Santa Cruz, Dallas, TX) was first concentrated to 50 µl agarose per 100 µl volume. 200 µl of plasma were then combined with 100 µl of concentrated protein A/G PLUS-agarose and incubated for 4 h at 4˚C under constant rotation. The mixture was then centrifuged for 5 min at 1000 x g. 200 µl of supernatant were again mixed with 100 µl of fresh protein A/G PLUS-agarose and subjected to overnight incubation at 4˚C under constant rotation. After two centrifugation steps (both 5 min, 1000 x g), the resultant cleared plasma was subjected to the ELISA procedure.

## Western blotting

Samples from Fab fragment generation (purified and unpurified fragments as well as undigested antibodies) and from Ig-cleared plasma (0.125 µl plasma per lane) were subjected to a standard sodium dodecyl sulfate polyacrylamide gel electrophoresis (SDS-PAGE) under reducing conditions. Proteins were transferred onto polyvinylidene fluoride (PVDF) membranes using the Trans-Blot® Turbo™ device (both Bio-Rad). Membranes were blocked overnight at 4˚C with 2–5% blotting grade blocker (Bio-Rad) plus 0.05% Tween 20 (Sigma-Aldrich) in PBS without calcium and magnesium. For samples from Fab fragment generation, immunoglobulins (or their respective fragments) were probed with an HRP-labeled goat anti-mouse IgG (H+L) polyclonal antibody (final concentration 2 ng/ml, RRID:AB_1185566, no. 32430, Thermo Fisher Scientific). For Ig-cleared plasma, immunoglobulins were detected with an HRP-labeled F(ab')₂ fragment goat anti-human IgG + IgM (H+L) polyclonal antibody (final concentration 16 ng/ml, RRID:AB_2337597, no. 109-036-127, Jackson Immuno Research, Ely, United Kingdom). After antibody incubation for 1 h at room temperature, signals were developed with ECL Ultra substrate (Luminogen, Southfield, MI) onto CL-XPosure™ Films (Thermo Fisher Scientific). Kaleidoscope™ Prestained Standard (Bio-Rad) or Sharp Prestained Protein Standard (Thermo Fisher Scientific) were used as molecular weight markers.

## Final (modified) MPO-DNA complex ELISA protocol

The final, optimized protocol for detection of MPO-DNA complexes holds one major modification as compared to the initial version. Calibrator and plasma samples were diluted into sample diluent (component of ABTS ELISA Buffer Kit) supplemented with 10% v/v TRU Block™ Ready. Calibrator was applied in a series of consecutive 1:2 dilution steps, starting with a dilution of 1:50, while plasma samples were prediluted at 1:100. For each sample, specific signals for MPO antibody and unspecific signals for isotype control coated wells were recorded in duplicates on the same plate. The MPO-DNA complex content of plasma samples was calculated as follows: after blank correction, the optical density obtained for the isotype control was subtracted from the optical density of MPO antibody-coated wells. This result was used to calculate the MPO-DNA complex concentration according to the calibration curve. To adjust for high interassay variation, three control plasma samples were applied to each plate in the endotoxemia and AAA sample analysis. The calculated MPO-DNA levels of the controls were set in reference to the initially determined values of the first plate. Thus, an averaged "conversion factor" was determined and subsequently applied to all plasma measurements of the respective plate.

## Experimental endotoxemia study

The study was approved by the institutional ethics committee of the Medical University of Vienna (approval number 1076/2016), was registered at EudraCT database (2016-000309-34) as well as clinicaltrials.gov (NCT02875028) and was conducted in 2016 at the Department of Clinical Pharmacology, Medical University of Vienna, adhering to The Code of Ethics of the World Medical Association (Declaration of Helsinki). All study participants gave written informed consent. Healthy volunteers > 18 years of age were administered a bolus infusion of lipopolysaccharide (LPS, 2 ng/kg body weight, Clinical Center Reference Endotoxin, National Institute of Health) [31]. While this study encompassed two treatment arms to evaluate the effects of voraxapar compared to placebo, only plasma samples from seven participants of the placebo group (receiving empty lactose-starch capsules 24 h prior to LPS infusion) were included in the present analysis. Peripheral venous blood anticoagulated with CTAD was collected immediately before (0 h) and 2 h, 4 h, 6 h, 8 h and 24 h after LPS administration.

## AAA study design

The study was approved by the institutional ethics committee of the Medical University of Vienna (approval number 1729/2014), was conducted according to The Code of Ethics of the World Medical Association (Declaration of Helsinki) and all study participants gave written informed consent. Forty advanced AAA patients prior to surgical repair were included between 2014 and 2016. Demographics were recorded by means of a structured questionnaire. All study participants were recruited at the Vascular Surgery outpatient clinic of the Vienna General Hospital and underwent a computed tomography angiography (CTA) scan. Exclusion criteria comprised recent (< 1 year) tumor incident and/or chemotherapy, systemic autoimmune or hematological disease and organ transplantation. Peripheral venous blood anticoagulated with citrate was collected upon study inclusion.

## Plasma preparation and analysis of additional blood parameters

Blood was processed within 60 minutes of withdrawal. Blood was centrifuged twice (1000 x g followed by 10000 x g, both for 10 min at 4˚C) to obtain platelet-free plasma and stored in aliquots at -80˚C. Plasma MPO (Human Myeloperoxidase Quantikine ELISA Kit, Bio-Techne)

and DNA-histone complexes (Cell Death Detection ELISA, Sigma-Aldrich) were quantified according to the manufacturer's recommendations. NE levels were assessed with the Human Sepsis Magnetic Bead Panel 3 (EMD Millipore) in a Luminex MagPix instrument (Thermo Fisher Scientific) for AAA study samples or with PMN (Neutrophil) Elastase Human ELISA Kit (Thermo Fisher Scientific) for endotoxemia plasma samples. Citrullinated histone H3 (citH3) was measured as reported elsewhere [32]. Differential blood count was determined with a Sysmex XN-350 device. Biochemical parameters were assessed by the Vienna General Hospital routine laboratory.

## Statistical analysis

With regard to ELISA optimization experiments, data are consistently shown as line graphs for the calibrator or bar diagrams for plasma samples. Optical densities are given unmodified, i.e. without subtracted blank values (which were additionally recorded). In each optimization experiment, blood donors are designated with Arabic numerals starting from "1". Concerning the observational AAA study, nominal variables are presented as counts and percent of sample group. Continuous variables are given as median and interquartile range (IQR) and have been assessed with Spearman's coefficient $r_s$ for correlations. Regarding the experimental endotoxemia study, data are depicted as medians and their 50% confidence intervals. Wilcoxon signed-rank test was applied to assess parameter deviations from baseline and Spearman's coefficient $r_s$ to evaluate correlations. Statistical testing was performed with SPSS Statistics 24.0 software (IBM, Armonk, NY) and was considered significant at a two-sided p value < 0.05. Raw data underlying the figures and tables of this manuscript are compiled in the S1 Data.

## Results

### Quantitation of *in vitro* generated NETs using the standard MPO-DNA ELISA

*In vitro* NETs can be triggered by activation of isolated neutrophils with stimuli, such as PMA or calcium ionophores. The supernatant of these reactions is then sampled for released DNA and neutrophil proteins, as performed by MPO-DNA complex ELISA. Furthermore, these *in vitro* samples are often applied as standards or calibrators when testing *in vivo* derived samples. Thus, to implement and test previously published MPO-DNA ELISA protocols [12, 28], we prepared three different batches of supernatant of PMA- or control-treated neutrophils from two different donors. They proved similar results regarding their content of MPO and DNA, when measured separately by MPO ELISA or PicoGreen dsDNA assay, respectively. Of note, stimulation with PMA resulted in a robust upregulation of MPO and DNA levels compared to control treatment. When the three supernatants of PMA-treated neutrophils were titrated in the MPO-DNA ELISA, they revealed comparable optical densities at 1:400 dilution, whereas the signals for the low (1:50) and high (1:1600) dilution of calibrator samples varied more extensively (Table 1).

Optical density values of blank wells generally ranged at 0.038 ± 0.013 (n = 17). Comparably, ELISA signals of control-treated neutrophils were close to background noise, i.e. MPO-DNA complexes could barely be detected (Fig 1A), although both, MPO and DNA were present in these preparations (Table 1).

Thus, we successfully implemented the previously published MPO-DNA ELISA protocol for *in vitro* generated NET samples. Of note, reducing the concentration of the coating antibody by half (to possibly save expenses) did not translate into a relevant change of the ELISA

**Table 1. MPO, DNA and MPO-DNA content of *in vitro* generated NET samples.**

| Batch | Neutrophil purity [%] | Treatment | MPO [ng/ml] | DNA [ng/ml] | MPO-DNA [optical density] | | | |
|---|---|---|---|---|---|---|---|---|
| | | | | | 1:50 dilution | 1:400 dilution | 1:1600 dilution | n |
| #1 | 94.0 | Control | 74.97 | 111.16 | 0.065 ± 0.002 | 0.040 ± 0.007 | 0.082 ± 0.001 | 2 |
| | | PMA | 226.02 | 646.95 | 2.743 ± 1.248 | 0.238 ± 0.102 | 0.082 ±0.026 | 5 |
| #2 | 95.8 | Control | 83.38 | 62.02 | 0.081 ± 0.017 | 0.051 ± 0.024 | 0.045 ± 0.019 | 3 |
| | | PMA | 214.74 | 645.16 | 2.410 ± 0.750 | 0.240 ± 0.071 | 0.078 ± 0.017 | 11 |
| #3 | 92.5 | Control | 45.40 | 70.30 | 0.058 ± 0.022 | 0.038 ± 0.007 | 0.036 ± 0.001 | 2 |
| | | PMA | 251.77 | 613.30 | 2.629 ± 0.755 | 0.234 ± 0.058 | 0.094 ± 0.036 | 6 |

DNA, deoxyribonucleic acid; MPO, myeloperoxidase; PMA, phorbol 12-myristate 13-acetate.

Neutrophils (obtained from two healthy donors: batches #1 and #2 from donor 1, batch #3 from donor 2) were stimulated with PMA or solvent control for 3 h and the resulting supernatants were assessed for MPO and DNA content separately, or by the initial MPO-DNA ELISA protocol for formed complexes. Optical density range is given for sample dilutions of 1:1600, 1:400 and 1:50, corresponding to relative concentrations of 1, 4 and 32. Mean ± SD refer to the maximum number (n) of assays for MPO-DNA complexes performed with the respective batch. Please refer to the S1 Table for data analysis restricted to control and PMA treated samples compared in the same assays.

signals for supernatants of PMA-treated neutrophils (S1A Fig). However, we continued using 5 µg/ml coating antibody to stay comparable with earlier studies [11, 28].

## Standard MPO-DNA ELISA yields specific signals for *in vitro* generated NETs but lacks specificity for NET detection in plasma samples

Next, we assessed whether the MPO-DNA complex ELISA demonstrates specificity for *in vitro* generated as well as *in vivo* occurring NETs. To this end, we compared microwells coated with MPO antibody, the corresponding isotype control or left the wells uncoated. The supernatant of PMA-stimulated neutrophils served as a source of *in vitro* generated NETs and as assay calibrator, while human plasma samples were the source of *in vivo* occurring NETs. We applied plasma anticoagulated with citrate or CTAD, as previously performed by others [20, 28]. The MPO capture antibody proved to be specific for *in vitro* generated NETs since signals for isotype control or uncoated wells were minimally above blank levels (blank values were 0.060, 0.053 and 0.047 for MPO antibody, isotype control and uncoated wells, respectively). The optical densities of MPO antibody-coated wells ranged between 3.235 and 0.099 for relative calibrator concentrations spanning from 32 to 1 (Fig 1B). For plasma samples the situation presented quite differently. While uncoated microwells gave little or no signal above blank levels, the optical densities for isotype control wells were similar to those detected for MPO antibody-coated wells (Fig 1C). This pointed to an unspecific reaction of plasma components with the applied coating antibody, while the blocking procedure seemed to be sufficient, as signals of uncoated wells were similar to the blank values. Of note, there was substantial variation among plasma donors regarding the level of unspecific signal which was, however, reproducible in relative terms when samples were repeatedly assayed on ELISA plates.

## Use of different antibodies does not eliminate unspecific reactions of plasma components

In an attempt to improve the ELISA by replacement of the original mouse monoclonal MPO coating antibody clone 4A4, we tested a rabbit polyclonal antibody previously described for detection of MPO-DNA complexes [24, 33]. However, calibrator (*in vitro* generated NETs) detection suffered from high background and a low dynamic range (Fig 2A). We then reversed the specificity of the antibody pair as practiced by Dicker *et al.* for NE-DNA complexes [25], using an antibody

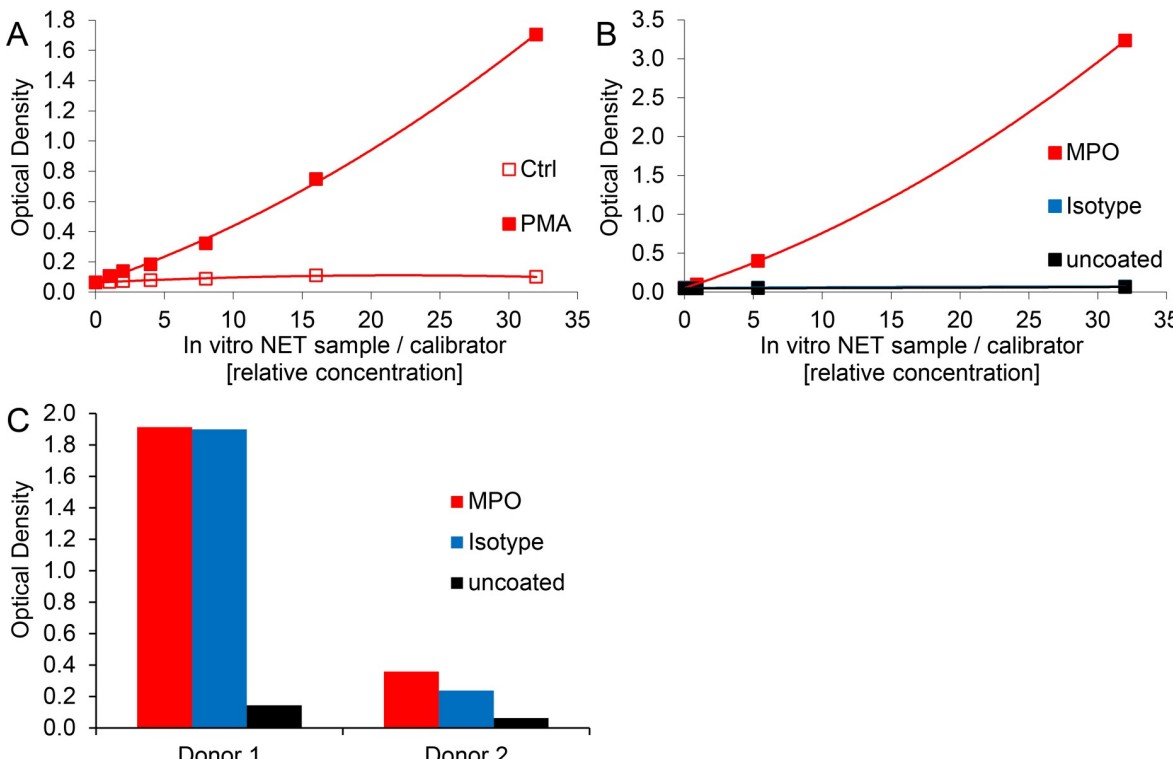

**Fig 1. The standard MPO-DNA complex ELISA protocol yields dose-dependent and specific signals for *in vitro* induced NETs but lacks specificity for NET detection in human plasma samples.** A) Supernatants of control- and PMA-treated, freshly isolated human neutrophils were diluted at 1:50, 100, 200, 400, 800 and 1600 (corresponding to relative concentrations of 32, 16, 8, 4, 2 and 1) and assessed by the initial MPO-DNA complex ELISA protocol. Shown data correspond to one representative experiment of calibrator batch #2. B) and C) Microwells were coated with MPO antibody, IgG2b isotype control or were left uncoated. B) Supernatant of freshly isolated and PMA-stimulated human neutrophils at dilutions corresponding to relative concentrations from 32 to 1 or C) plasma samples from two human donors at 1:2 dilution were evaluated with the initial ELISA protocol. Note that data points of isotype control and uncoated wells overlap in 1B.

directed against DNA to capture MPO-DNA complexes and an HRP-labeled antibody recognizing MPO for detection. Again, the range of calibrator signal was modest and increasing concentrations of detection antibody resulted in an elevated background (Fig 2B). When MPO detection was changed to a biotinylated anti-MPO antibody and streptavidin-HRP secondary reagent, calibrator samples yielded signals comparable to the initial MPO-DNA complex ELISA protocol (Fig 2C). Thus, *in vitro* generated NETs were also efficiently quantified by the inverse DNA-MPO ELISA setup and showed no unspecific reaction with isotype control or uncoated wells. In contrast, assessment of plasma samples did not improve. Signals for isotype control and even for uncoated microwells were similar to those for the specific anti-ds DNA coating antibody (Fig 2D). These unspecific reactions of plasma components observed for uncoated microwells were mainly attributed to the biotinylated antibody targeting MPO (S1B Fig). We therefore decided to continue with the initial MPO-DNA complex ELISA protocol and to attempt further steps of improvement.

## Unspecific plasma signals in the MPO-DNA ELISA are not due to aberrant MPO capturing or peroxidase activity

We first compared the efficiency of MPO capturing in the initial ELISA protocol, i.e. calibrator and plasma samples were incubated in microwells coated with MPO or isotype antibody. The unbound material was then retrieved and subjected to MPO quantification by standard MPO

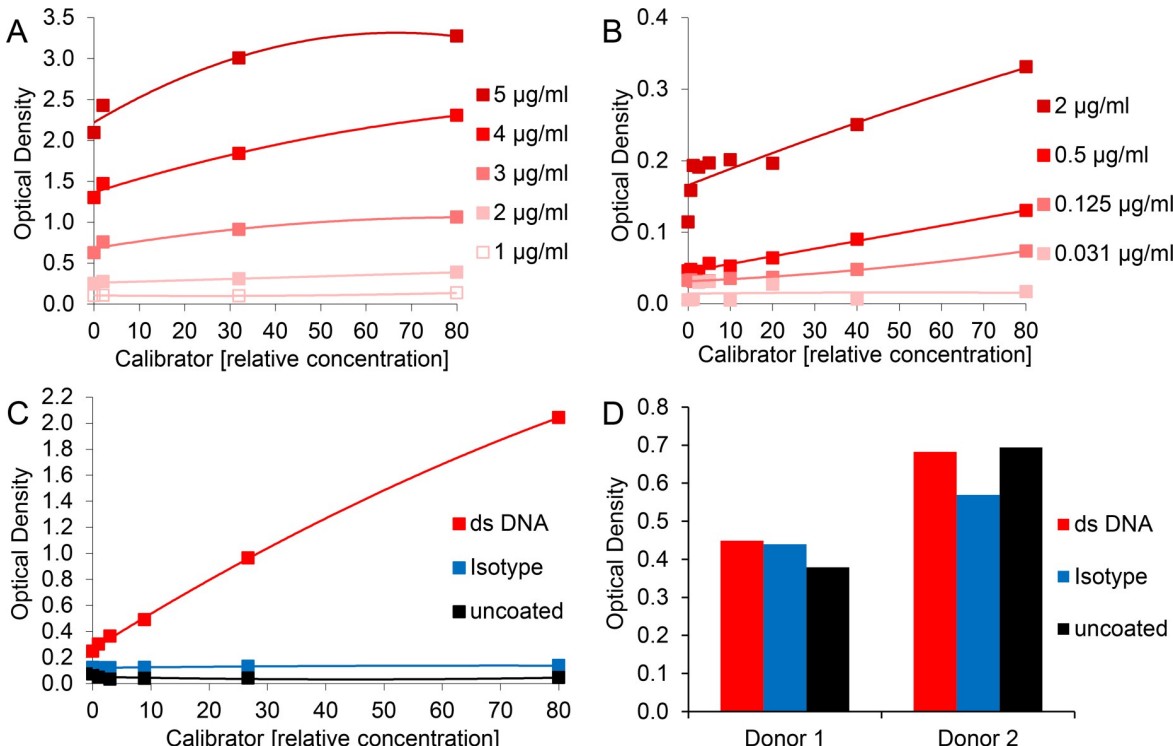

**Fig 2. Exchange of assay antibodies does not eliminate unspecific reactions of plasma components.** For assessment of A)—C) serially diluted calibrator and D) 1:2 diluted plasma, the following modifications of the initial MPO-DNA complex ELISA protocol were applied: Microwells were coated with A) 1 - 5 μg/ml rabbit anti-human MPO polyclonal antibody (no. 07-496-I, EMD Millipore), B) 4 μg/ml mouse anti-ds DNA monoclonal antibody (no. ab27156, Abcam) or C) and D) 1 μg/ml mouse anti-ds DNA monoclonal antibody (no. ab27156, Abcam), 1 μg/ml mouse IgG2a kappa monoclonal isotype control (no. 010-001-332, Rockland Immunochemicals) or were left uncoated. For MPO-DNA complex detection A) peroxidase-conjugated mouse anti-DNA monoclonal antibody (clone MCA-33, as described for the initial ELISA protocol), B) 0.031 – 2 μg/ml HRP-labeled mouse anti-human MPO monoclonal antibody (no. NBP2-41406H, Bio-Techne) or C) and D) 0.5 μg/ml biotinylated mouse anti-human MPO monoclonal antibody (no. HM2164BT, HyCult Biotech) followed by 0.5 μg/ml streptavidin-HRP were applied.

ELISA and compared to the MPO content of the starting material. MPO capturing was clearly less efficient for plasma than for *in vitro* generated (calibrator) NETs despite comparable MPO starting concentrations for samples applied to the microwells (3.52 ng/ml, 7.60 ng/ml and 4.23 ng/ml for donor 1, donor 2 and calibrator at a relative concentration of 32, respectively). The isotype control antibody did not remove substantial amounts of the protein (Fig 3A). Hence, binding of plasma MPO to the isotype cannot account for the unspecific signal in MPO-DNA ELISA. As MPO can potentially process the HRP substrates ABTS and 3,3′,5,5′-tetramethylbenzidine (TMB) [34], we were concerned that captured sample MPO could interfere with the applied peroxidase-labeled detection antibody in signal generation. However, the captured MPO protein did not interfere, since omission of the secondary antibody completely abrogated substrate turnover when calibrator was applied (Fig 3B). A similar pattern was found for 1:2 diluted plasma samples of two donors: optical densities ranged at 4.816 and 0.168 after application of peroxidase-labeled detection antibody and at 0.053 and 0.057 when the detection antibody was omitted.

## Exchange of ELISA ancillary components does not alter unspecific signals of plasma samples

We next assessed whether the application of different ELISA ancillary components could help to enhance assay specificity for plasma MPO-DNA complexes. We first exchanged blocking

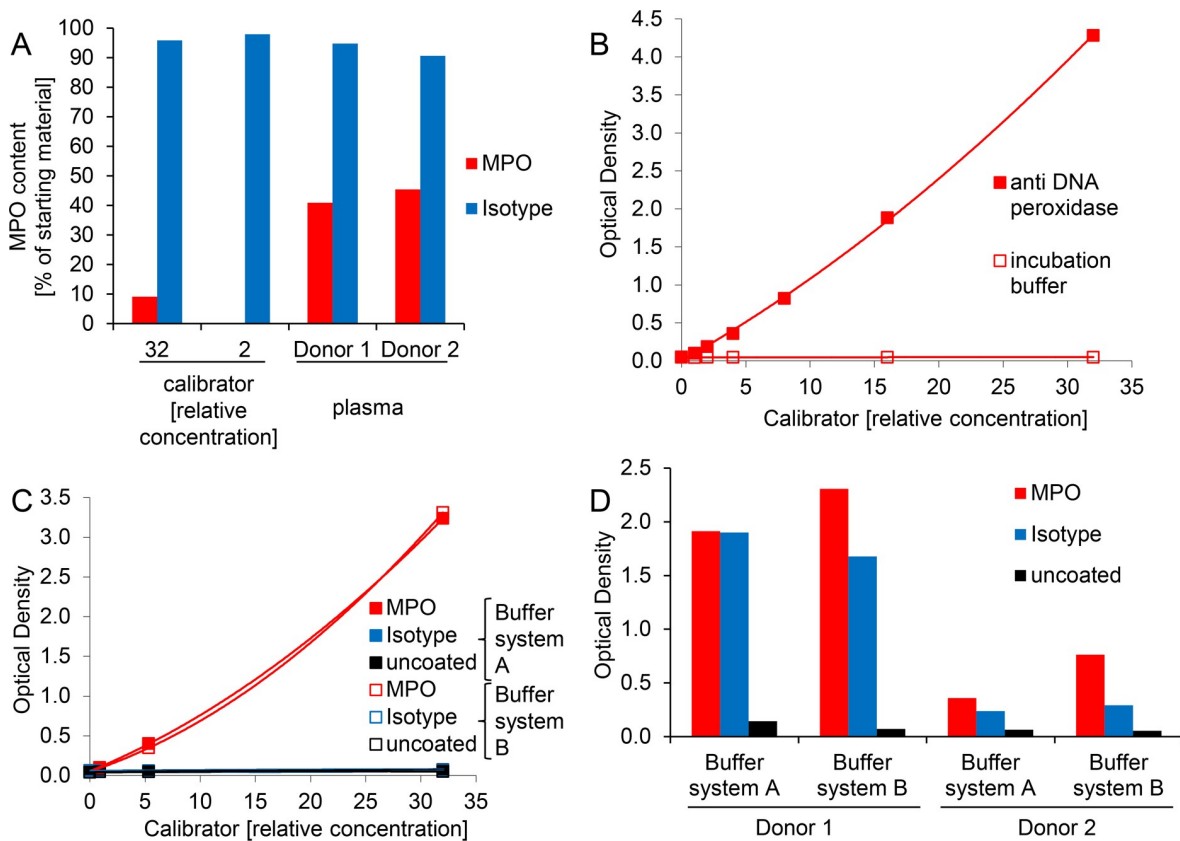

**Fig 3. Unspecific plasma signals in MPO-DNA ELISA are not due to aberrant MPO capturing or peroxidase activity and are not altered by exchange of ELISA ancillary components.** A) Calibrator at two different dilutions or 1:2 diluted plasma was incubated in microwells coated with MPO or isotype antibody using the initial MPO-DNA complex ELISA protocol. The unbound material was then retrieved from the wells and subjected to MPO quantification (using a commercial MPO ELISA kit) and expressed as protein content in proportion to the starting material. B) Serially diluted calibrator was incubated in microwells coated with MPO antibody using the initial MPO-DNA complex ELISA protocol. Substrate turnover was monitored after application of peroxidase-labeled detection antibody or of plain dilution buffer (incubation buffer) to reveal potential substrate reactivity of captured MPO. C) and D) Microwells were coated with MPO antibody, isotype control or were left uncoated and incubated with C) calibrator or D) 1:2 diluted plasma samples. Samples designated "buffer system A" were processed as described for the initial MPO-DNA complex ELISA protocol, i.e. with ancillary components of the ABTS ELISA Buffer Kit. For samples labeled "buffer system B" the following reagents were exchanged: blocking buffer and sample diluent were substituted with incubation buffer (component number 5 of Cell Death Detection ELISA), while wash buffer was exchanged to component number 4 of Cell Death Detection ELISA. Note that data points of buffer systems A and B overlap in 3C, i.e. the data points for MPO antibody coated wells in buffer system B (red open squares) are partly covered by the indicated measurements for MPO antibody coated wells in buffer system A (red filled squares). Similarly, signals for isotype control and uncoated wells in buffer system B (blue and black open squares, respectively) as well as for isotype control in buffer system A (blue filled squares) are masked by the indicated measurements of uncoated wells in buffer system A (black filled squares).

buffer, sample diluent and wash buffer of the ABTS ELISA Buffer Kit (Buffer system A, Pepro Tech) to comparable reagents provided with the Cell Death Detection ELISA (Buffer system B, Sigma Aldrich), which also contains the peroxidase-conjugated anti-DNA antibody utilized for detection of MPO-DNA complexes. No substantial differences were observed, i.e. both ELISA variants were suited to quantitate *in vitro* generated calibrator NETs (Fig 3C) but lacked specificity for MPO-DNA complexes in plasma samples (Fig 3D). We then replaced the blocking buffer, sample diluent and detection antibody diluent of the initial MPO-DNA complex ELISA protocol by an in-house blocking buffer. Again, we did not find any improvement for the analysis of plasma samples but rather elevated blank signals and a reduced dynamic range of the calibrator (S2A and S2B Fig). Various other tested buffer formulations optimized to

reduce unspecific cross-reactions and matrix effects (termed LowCross-Buffers®) [35, 36] were not beneficial in our setting, since signals of both, calibrator and plasma samples were reduced to a minimum (S2C and S2D Fig). This suggests that complexes of MPO and DNA were dismantled and/or binding of the assay antibodies to MPO-DNA complexes was reduced in these buffer systems. Similar results were found for application of a commercially available blocker (The Blocking Solution), yielding significantly lower signals compared with the initial ELISA protocol, without improving ELISA specificity (S2E and S2F Fig).

## Cross-reactivity of plasma components in MPO-DNA ELISA is observed for various mouse IgG isotypes

We next assessed whether the unspecific signals observed for plasma samples in MPO-DNA ELISA were related to a particular antibody isotype or manufacturer. Neither exchange to a different commercial provider nor to another class of mouse IgG resolved the issue (Fig 4A). To further investigate the possibility that the observed interferences for plasma were mediated via the Fc portion of the coating antibody, we processed the MPO antibody as well as the iso-type control to Fab and Fc fragments. Western blot analysis confirmed that the immunoglobu-lins were completely processed to their respective fragments and that Fab fragments were successfully purified (Fig 4B and 4C). The purified Fab fragments (fraction 1 of both, MPO antibody and isotype control) were then applied in the ELISA procedure. Calibrator detection was comparable to the ELISAs previously performed with full length IgG (Fig 4D). Again, for plasma samples no significant change was achieved since the isotype control showed substan-tial background signal (Fig 4E). Thus, cross-reactions of plasma components were not or just in part mediated via the Fc portion of the coating antibody.

## DNA degradation abolishes detection of *in vitro* generated MPO-DNA complexes but enhances unspecific plasma signals

We hypothesized that DNA might contribute to the unspecific plasma reactions and therefore DNA digestion should eliminate both, the specific MPO-DNA ELISA signal as well as the aberrant reaction with isotype control. Thus, we incubated calibrator and plasma samples with an excess of MNase and then subjected the resulting material to the ELISA procedure. For cali-brator samples, DNA digestion essentially abolished the observed signal for MPO antibody-coated microwells and left the signal for isotype control or uncoated wells unaffected (Fig 5A). Regarding plasma samples, MNase digestion increased rather than decreased the signals for both, MPO antibody and isotype coated wells, again giving comparable optical densities (Fig 5B and 5C). Background noise of uncoated wells was unaffected (Fig 5C).

## Detection of *in vitro* generated MPO-DNA complexes is abrogated when spiked into plasma

We next determined if we could detect *in vitro* generated MPO-DNA complexes that were spiked at increasing concentrations into 1:4 or 1:100 pre-diluted plasma. The optical densities of plasma spiked with different amounts of MPO-DNA complexes equaled signals for 1:4 diluted plasma without spiking (Fig 6A). At a higher 1:100 dilution of plasma the addition of *in vitro* prepared MPO-DNA complexes translated into increasing ELISA signals, but with a limited and donor-dependent dynamic range. Furthermore, signals for isotype control coated wells frequently exceeded those for MPO coated wells (Fig 6B).

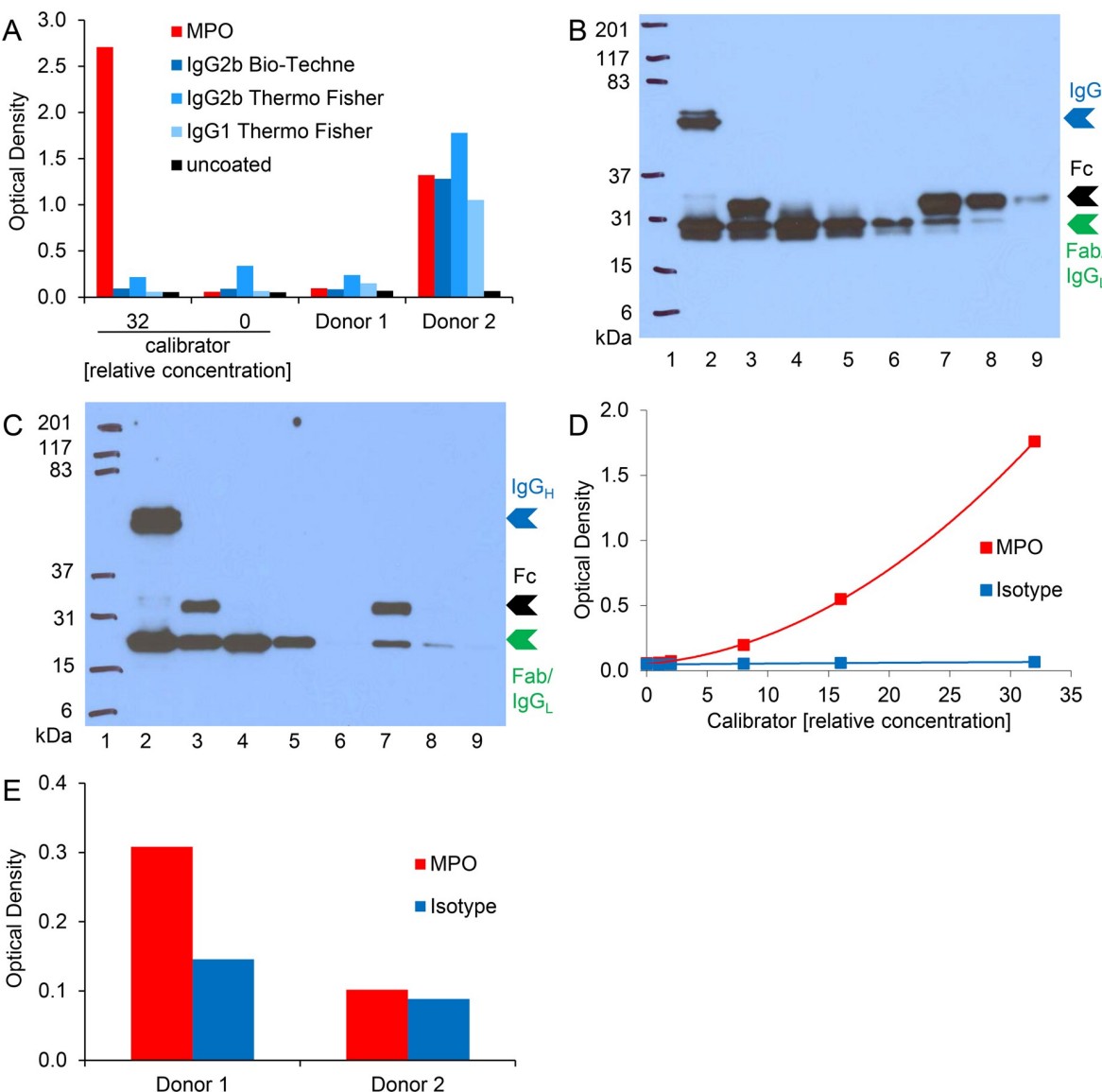

**Fig 4. Cross-reactivity of plasma components in MPO-DNA ELISA is observed for various mouse IgG isotypes.** A) Microwells were coated with MPO antibody, various isotype controls or were left uncoated. Calibrator or 1:2 diluted plasma samples were analyzed according to the initial MPO-DNA complex ELISA protocol. The calibrator with a relative concentration of "0" equals blank. B) MPO antibody (Bio-Rad) and C) isotype control (IgG2b, Bio-Techne) were processed to Fab and Fc fragments. Western blot analysis after reducing SDS-PAGE (12.5% gel) of unprocessed antibodies (lane 2), papain cleaved unpurified antibodies (lane 3), purified Fab fragments (fractions 1–3, lanes 4–6) as well as Fc fragments (fractions 1–3, lanes 7–9) are depicted. Molecular weight marker (Kaleidoscope™ Prestained Standard) was applied in lane 1 and is given in kDa. Intact immunoglobulin G heavy chains (IgG$_H$) in lane 2 and the respective Fc and Fab fragment chains in lanes 3–9 are indicated with blue, black and green arrowheads, respectively. Please note that the immunoglobulin G light chain (IgG$_L$) remains unaltered by the fragmentation process and has a molecular weight close to the fragmented Fab heavy chain. Hence, the two protein bands could not be resolved and are both indicated by the green arrowhead. D) and E) MPO antibody and isotype control (for both: Fab fragment, fraction 1) were coated onto microwells and incubated with D) serially diluted calibrator or E) 1:2 diluted plasma samples. The further ELISA procedure was performed as indicated for the initial MPO-DNA complex ELISA protocol.

## Plasma dilution does not increase the specificity of MPO-DNA detection in plasma samples

We studied the effect of plasma dilutions on the specificity of MPO-DNA complex detection in more detail. Plasma dilutions between 1:2 and 1:10 did not entail a substantial signal change

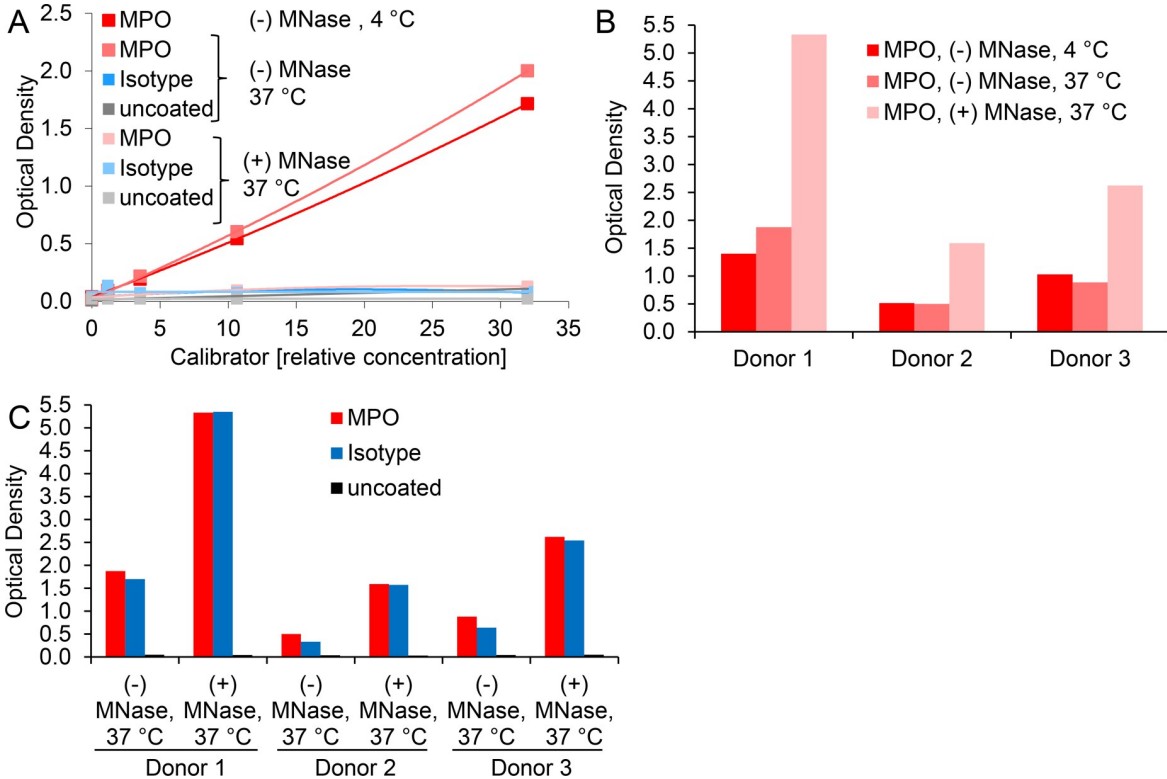

**Fig 5. DNA degradation abolishes detection of *in vitro* generated MPO-DNA complexes but enhances unspecific plasma signals.** A) Calibrator or B) and C) plasma DNA was digested with (+) A) 2 U/ml or B) and C) 20 U/ml MNase at 37°C for 60 min. Control samples were incubated at 37°C or kept on ice (4°C), both without (-) MNase supplementation. Microwells were A) and C) coated with MPO antibody, isotype control or were left uncoated or B) coated with MPO antibody only. The further ELISA procedure was performed as outlined for the initial MPO-DNA complex ELISA protocol. Plasma was applied at a final dilution of 1:2. Donors 1–3 are identical in B) and C).

(Fig 7A). Surprisingly, a moderate elevation of optical densities was observed with higher plasma dilutions in the range between 1:30 and 1:100 (Fig 7B). With further increase of plasma dilution, ELISA signals dropped again (Fig 7C). Of note, none of the applied plasma dilutions

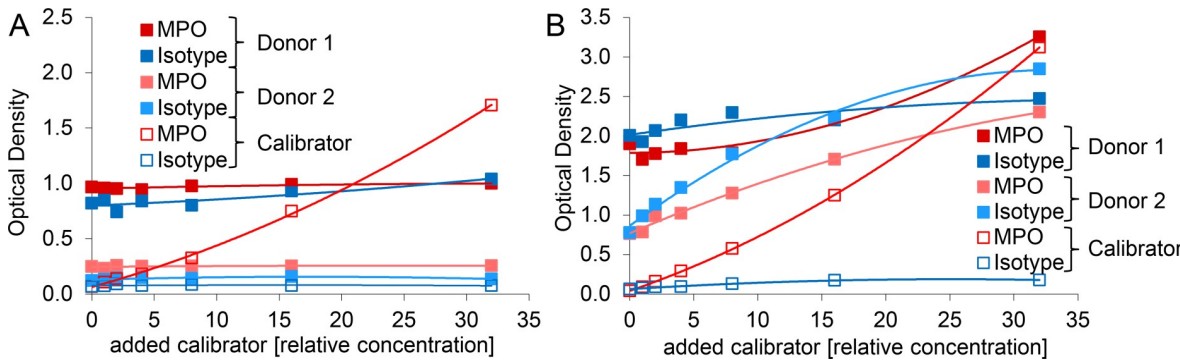

**Fig 6. Detection of *in vitro* generated MPO-DNA complexes is abrogated when spiked into plasma.** Microwells were coated with MPO antibody or isotype control. Increasing amounts of calibrator (relative concentration of supernatant of PMA stimulated neutrophils) were added to plasma of two distinct donors which was A) 1:4 or B) 1:100 pre-diluted with sample diluent. Calibrator without plasma was additionally measured (open symbols). Samples were further processed according to the initial MPO-DNA complex ELISA protocol. Note that an added relative calibrator concentration of "0" represents plain plasma at the respective dilution.

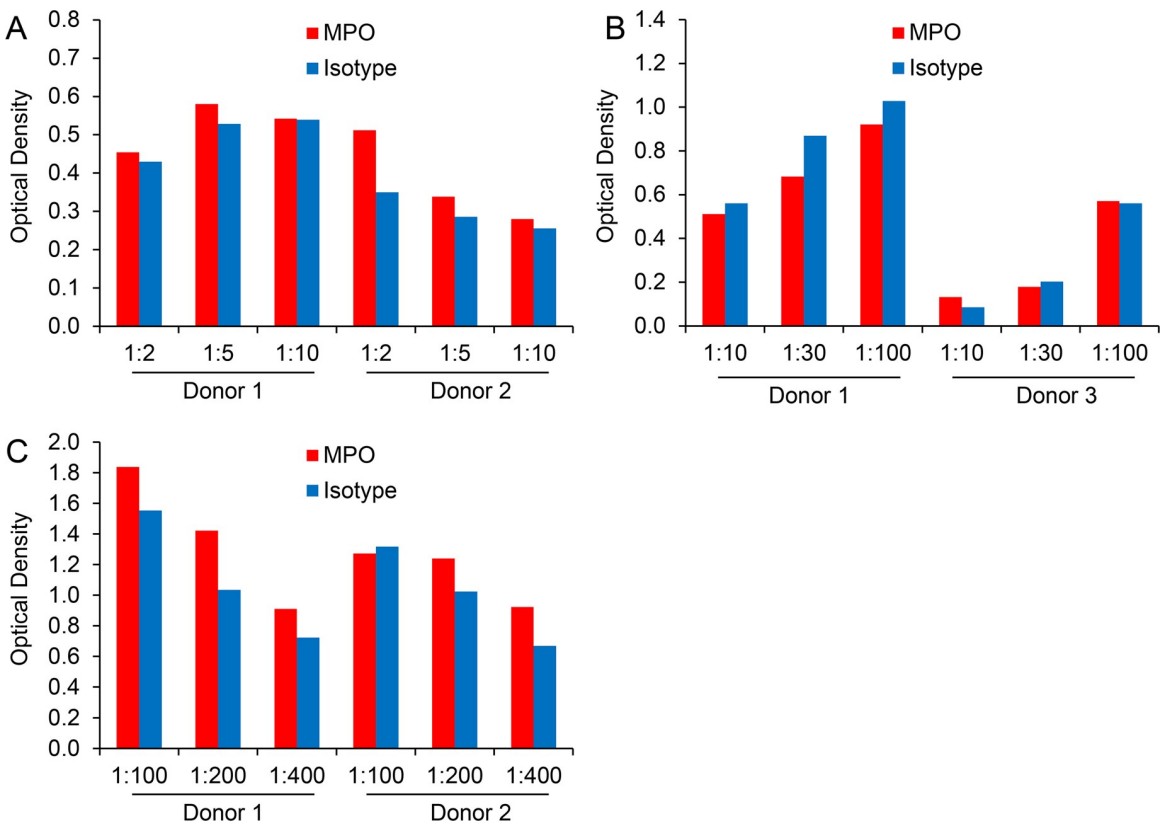

**Fig 7. Plasma dilution does not increase the specificity of MPO-DNA detection in plasma samples.** Microwells were coated with MPO antibody or isotype control. Plasma was applied at dilutions of A) 1:2, 1:5 or 1:10, B) 1:10, 1:30 or 1:100 and C) 1:100, 1:200 or 1:400 and further processed according to the initial MPO-DNA complex ELISA protocol.

significantly improved the specificity of the MPO-DNA complex detection when compared to isotype control.

## Establishment of a modified MPO-DNA ELISA for detection of MPO-DNA complexes in human plasma

Multivalent substances, such as heterophile antibodies present in patient specimens can provoke immunoassay signals independent from the analyte by simply bridging the applied assay antibodies. Similar effects can be triggered by human anti-animal antibodies (HAAAs) comprising the subgroup of human anti-mouse antibodies (HAMAs) [37–39]. We therefore tried to pre-adsorb any potential interfering substance on a microplate coated with isotype antibody before submitting the retrieved (potentially "cleared") plasma sample to the ELISA procedure. This process did not sufficiently remove the interfering agent(s) and left the assay unchanged in terms of specificity (S3A Fig). In a second approach, we removed plasma immunoglobulins with protein A/G-coated agarose in advance to the ELISA procedure. Protein A/G has a strong affinity for IgG, but exerts a weak binding to IgM and IgA [40]. Western blotting using a detection antibody raised against human IgG and IgM revealed that both immunoglobulins were efficiently removed from plasma (S3B Fig). The effect of immunoglobulin depletion on the ELISA signal was donor-dependent. Elevated or diminished signals compared to plasma without Ig clearance were observed, without increasing the specificity of MPO-DNA detection in relation to isotype control (S3C Fig).

Commercially available immunoassays often contain specific blocking agents to prevent erroneous results [38, 41]. When the assay is based on capture or detection antibodies raised in mice, the addition of mouse serum or purified mouse immunoglobulins to human samples is suggested to divert the interfering substances [39, 42, 43]. Therefore, we further attempted to improve the MPO-DNA ELISA by adding graded doses of mouse IgG to calibrator and plasma samples. With increasing amounts of mouse IgG, *in vitro* generated NETs showed a trend to lowered signals for wells coated with MPO antibody and to elevated signals for wells coated with isotype control (Fig 8A). Supplementation of purified mouse IgG to human plasma

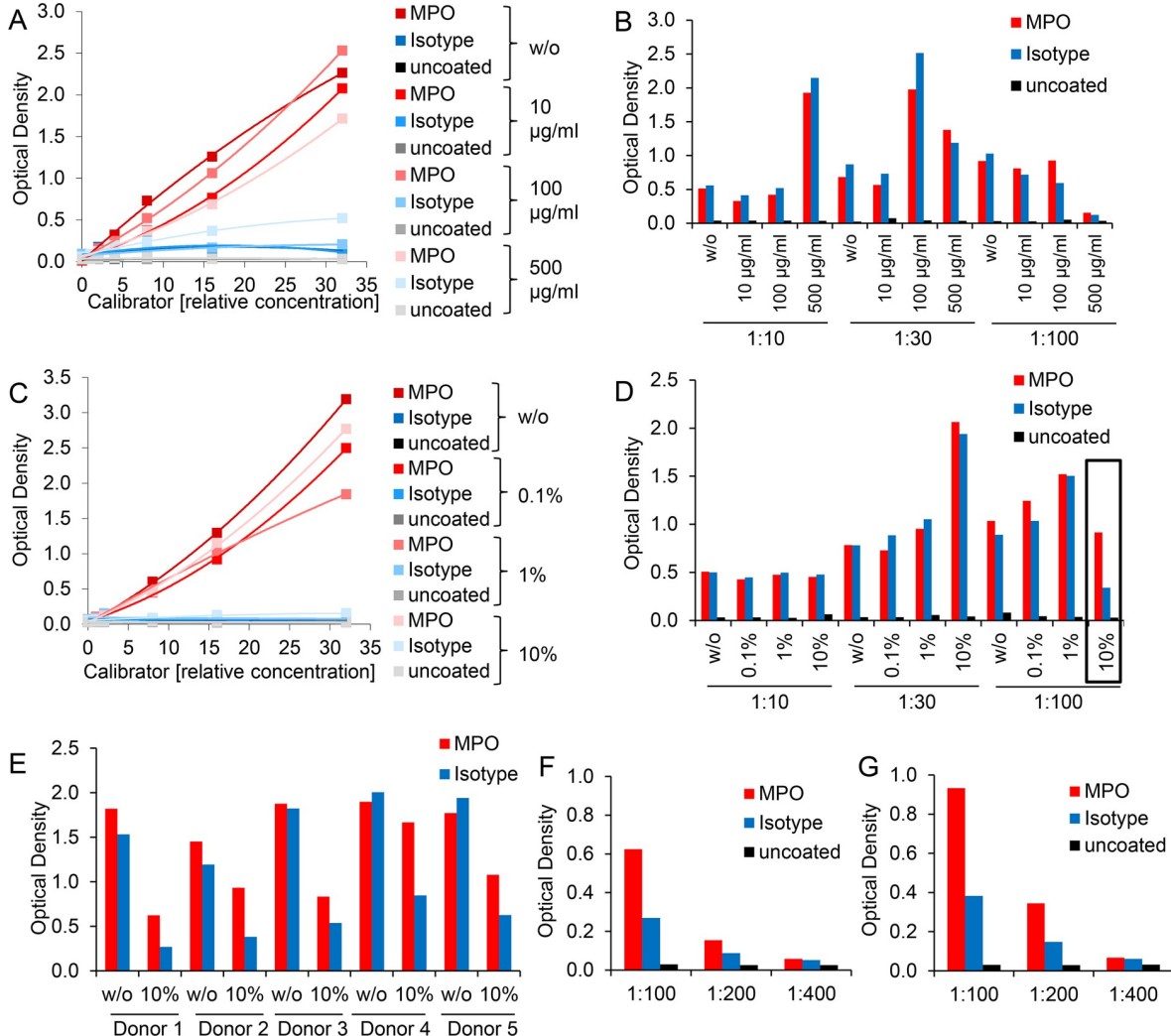

**Fig 8. Approaches to block multivalent interfering substances or HAMAs may improve the ELISA selectivity for MPO-DNA complexes in human plasma.** A) and C) Serially diluted calibrator or B) and D) plasma samples diluted at 1:10, 1:30 and 1:100 (with sample diluent of the ABTS ELISA Buffer Kit) were supplemented A) and B) without (w/o) or with increasing doses (10 μg/ml, 100 μg/ml, 500 μg/ml) of protein A-purified mouse IgG from normal mouse serum or C) and D) without (w/o) or with graded doses (0.1%, 1%, 10%) of TRU Block Ready reagent. The further procedure of the MPO-DNA complex ELISA was performed according the initial protocol in microwells coated with MPO antibody, isotype control or left uncoated. Note that data points of uncoated microwells overlap in 8A and C. E) Plasma was diluted 1:100 without (w/o) or with supplementation of 10% TRU Block Ready reagent. The further procedure of the MPO-DNA complex ELISA was performed according to the initial protocol in microwells coated with MPO antibody or isotype control. F) and G) Plasma of two donors at dilutions of 1:100, 1:200 and 1:400 was supplemented with 10% TRU Block Ready reagent. The further procedure of the MPO-DNA complex ELISA was based on the initial protocol with microwells coated with MPO antibody or isotype control or left uncoated.

enhanced ELISA signals depending on the plasma dilution: the more diluted the plasma, the less mouse IgG was required to achieve higher optical densities, but again without substantially improving ELISA specificity of MPO antibody versus isotype control signals (Fig 8B).

Thus, we further tested a commercially available immunoassay reagent, TRU Block Ready, which is designated to block HAMAs as well as heterophile antibody interference. Similar to the results for supplementation with mouse IgG, the incorporation of TRU Block Ready influenced calibrator performance towards lower optical densities but without substantially elevating background signals of isotype control coated or uncoated wells (Fig 8C). For plasma samples, addition of this reagent increased ELISA signals, in particular at 1:30 plasma dilution. Again, the impact of TRU Block Ready regarding the selectivity for MPO-DNA complexes in human plasma was small. However, for highly (1:100) diluted plasma an enlarged window between the signals for MPO antibody and isotype control was achieved, when plasma was supplemented with the highest dose of 10% v/v TRU Block Ready (Fig 8D). This observation could be reproduced in subsequent experiments with additional plasma donors (Fig 8E). Signal was still detectable at 1:200 plasma dilution, but reached the detection limit at 1:400 dilution (Fig 8F and 8G).

Based on all attempts to improve the applicability of the MPO-DNA ELISA for human plasma, we finally concluded to I) apply plasma at a dilution of 1:100, II) to supplement the sample diluent with 10% v/v TRU Block Ready and III) to consistently include isotype control coated wells in the ELISA measurements to be able to deduct unspecific from specific signals. With the aim of determining the intra- and interassay coefficients of variation (CV) for this modified MPO-DNA complex ELISA protocol six calibrator concentrations and three plasma samples were applied in duplicates to six consecutive ELISA plates. The mean intra- and interassay CV values for calibrator samples ranged at 9.3% and 12.7%, whereas plasma samples exhibited coefficients of 16.0% and 29.0%, respectively.

### Several markers of neutrophil activation and NET formation reflect *in vivo* NET induction but do not correlate with plasma levels of MPO-DNA complexes as determined by ELISA

To assess whether the modified MPO-DNA complex ELISA protocol is suited to monitor *in vivo* NET formation, plasma samples from a human endotoxemia study were investigated. In this experimental model of acute inflammation, seven healthy volunteers received low-dose LPS and blood was sampled at several time points from baseline to 24 hours. In addition to performing the modified MPO-DNA complex ELISA protocol, plasma samples were assayed (at 1:2 dilution) with the initial ELISA variant for comparison. Furthermore, we applied commercially available tests to determine plasma levels of MPO, NE and DNA-histone complexes and conducted a previously published immunoassay for citrullinated histone H3 (citH3) [32]. After a single bolus LPS infusion, the healthy volunteers displayed significantly and highly elevated plasma levels of NE, MPO, citH3 and DNA-histone (Fig 9A and 9B) which peaked at 4 h after LPS infusion [median (IQR) NE: 64 (45) ng/ml; MPO: 19 (9) ng/ml; citH3: 1516 (2105) ng/ml; DNA-histone: 123 (92) RU]. In contrast, MPO-DNA complexes decreased compared to baseline values according to the initial as well as final ELISA protocol (Fig 9C). This drop in measured plasma levels was observed regardless whether unspecific isotype signals were disregarded or were deducted from recorded MPO-DNA values before or after calculation of relative units (S4A–S4C Fig).

While all other markers of neutrophil activation (NE, MPO) and NET formation (citH3, DNA-histone) correlated significantly and positively, MPO-DNA complexes as assayed by the initial ELISA protocol displayed a negative correlation with NE and MPO (Table 2).

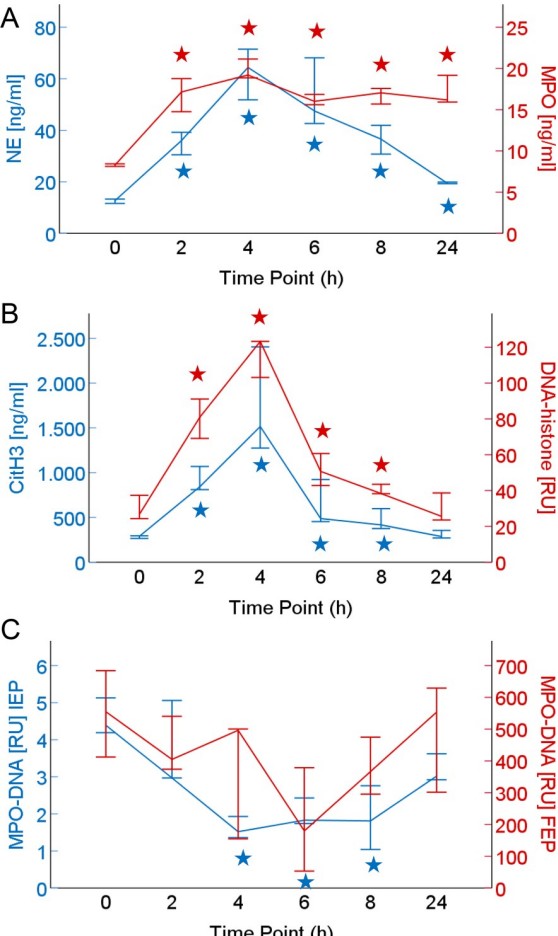

**Fig 9. Levels of MPO-DNA complexes and parameters of neutrophil activation and NET formation in plasma samples of an experimental endotoxemia model.** Blood of seven healthy volunteers was sampled at baseline (0 h) and at various time points after LPS infusion. Plasma was assessed for A) NE and MPO, B) citH3 and DNA-histone complexes and C) MPO-DNA complexes according to the initial (IEP) and the final ELISA protocol (FEP), both with isotype signals deducted before calculation of relative units (RU). Median values and their 50% confidence intervals are indicated. ★ p < 0.05 according to Wilcoxon signed-rank test (in comparison to baseline).

Similarly, when analyzing a set of plasma samples from 40 AAA patients (S2 and S3 Tables) which we have previously shown [44] to exhibit a moderately but significantly elevated level of the NET parameter citH3 when compared to healthy controls [median (IQR) AAA: 362 (203) ng/ml; healthy controls: 304 (112) ng/ml, p = 0.004], the plasma content of citH3 correlated significantly with MPO and NE (Table 3). However, none of the other markers of neutrophil activation or NET formation showed a correlation with plasma levels of MPO-DNA complexes as evaluated by the modified ELISA protocol.

## Discussion

In recent years, assessment of MPO-DNA complexes has been frequently applied to investigate NET formation with *in vitro* [17, 22, 24, 45, 46] as well as *in vivo* [17, 20, 24, 28] samples and findings are included in high impact publications [11, 12, 47], none of which included isotype control antibodies. Mostly, a classical sandwich ELISA format using a capture and a labeled detection antibody was used. Alternatively, a few research groups detected DNA attached to

**Table 2. Correlations between plasma levels of MPO-DNA complexes and parameters of neutrophil activation and NET formation in an experimental human endotoxemia model.**

| Parameter | $r_s$ | p-value | n |
|---|---|---|---|
| *MPO-DNA [RU]–FEP* | | | |
| MPO-DNA [RU]–IEP | 0.200 | 0.203 | 42 |
| CitH3 [ng/ml] | 0.104 | 0.511 | 42 |
| DNA-histone [RU] | 0.120 | 0.447 | 42 |
| NE [ng/ml] | -0.128 | 0.419 | 42 |
| MPO [ng/ml] | -0.126 | 0.427 | 42 |
| *MPO-DNA [RU]–IEP* | | | |
| CitH3 [ng/ml] | -0.231 | 0.142 | 42 |
| DNA-histone [RU] | -0.128 | 0.421 | 42 |
| NE [ng/ml] | **-0.450** | **0.003** | **42** |
| MPO [ng/ml] | **-0.406** | **0.008** | **42** |
| *CitH3 [ng/ml]* | | | |
| DNA-histone [RU] | **0.818** | **< 0.001** | **42** |
| NE [ng/ml] | **0.731** | **< 0.001** | **42** |
| MPO [ng/ml] | **0.501** | **0.001** | **42** |
| *DNA-histone [RU]* | | | |
| NE [ng/ml] | **0.554** | **< 0.001** | **42** |
| MPO [ng/ml] | **0.464** | **0.002** | **42** |
| *NE [ng/ml]* | | | |
| MPO [ng/ml] | **0.586** | **< 0.001** | **42** |

CitH3, citrullinated histone H3; DNA, deoxyribonucleic acid; FEP, final ELISA protocol; IEP, initial ELISA protocol; MPO, myeloperoxidase; NE, neutrophil elastase; $r_s$, Spearman's coefficient of correlation; RU, relative units. CTAD plasma samples of seven healthy volunteers at baseline and at various time points after LPS administration were assessed for MPO-DNA complexes (by the initial as well as the final ELISA protocol), NE, MPO, DNA-histone complexes and citH3. Correlation between parameters is expressed by Spearman's coefficient.

MPO by means of a fluorescent dye [48, 49] which, in our hands, was not sufficiently sensitive for MPO-DNA detection in human plasma. Most articles give limited information on the MPO-DNA ELISA setup and controls, in particular concerning clinical samples. When we tried to implement published ELISA protocols for MPO-DNA complex analysis in human plasma, we included an isotype control for the specific MPO coating antibody. Remarkably, we noticed high specificity and sensitivity for *in vitro* generated NET samples, while the MPO-DNA specificity for *in vivo* retrieved plasma samples was poor. We undertook substantial efforts to improve assay specificity for *in vivo* material which resulted in a modified ELISA protocol. Yet, measured MPO-DNA complexes in human plasma samples did not correlate with any other parameter of neutrophil activation or NET formation we compared. We would thus like to alert the NET research community that ELISA-based quantitation of MPO-DNA complexes in plasma samples is highly error-prone and that previously published data likely suffer from the discovered artefacts and should be interpreted with caution.

Blood markers which specifically reflect local processes of *in vivo* NET formation are sparse, as most parameters are detectable during neutrophil activation as well as NET formation and are hence not suited to differentiate between these processes. Apart from citrullinated histones [8, 32], DNA in complex with neutrophil-derived proteins such as NE or MPO has been proposed as specific NET marker. Yet it has to be considered that MPO-DNA complexes could potentially form when molecules interact in circulation. Since MPO is a positively charged

**Table 3. Correlations between MPO-DNA complexes and parameters of neutrophil activation and NET formation in plasma of AAA patients.**

| Parameter | $r_s$ | p-value | n |
|---|---|---|---|
| *MPO-DNA [RU]–FEP* | | | |
| CitH3 [ng/ml] | -0.059 | 0.716 | 40 |
| DNA-histone [RU] | 0.089 | 0.584 | 40 |
| NE [ng/ml] | 0.064 | 0.700 | 39 |
| MPO [ng/ml] | 0.043 | 0.791 | 40 |
| *CitH3 [ng/ml]* | | | |
| DNA-histone [RU] | 0.011 | 0.944 | 40 |
| NE [ng/ml] | **0.375** | **0.019** | **39** |
| MPO [ng/ml] | **0.650** | **< 0.001** | **40** |
| *DNA-histone [RU]* | | | |
| NE [ng/ml] | 0.048 | 0.770 | 39 |
| MPO [ng/ml] | 0.110 | 0.499 | 40 |
| *NE [ng/ml]* | | | |
| MPO [ng/ml] | 0.314 | 0.052 | 39 |

CitH3, citrullinated histone H3; DNA, deoxyribonucleic acid; FEP, final ELISA protocol; MPO, myeloperoxidase; NE, neutrophil elastase; $r_s$, Spearman's coefficient of correlation; RU, relative units.

Citrate plasma samples of AAA patients were assessed for MPO-DNA complexes (by modified ELISA protocol) and for NE, MPO, DNA-histone complexes and citH3. Correlation between parameters is expressed by Spearman's coefficient.

protein [50] it can bind to the negatively charged DNA as has previously been reported [51]. However, it was found that supernatant of PMA-treated "NETosing" neutrophils in contrast to apoptotic or necrotic neutrophils contains significant amounts of MPO-DNA complexes [24, 46]. Our own observations would corroborate the notion that MPO-DNA complexes are specifically formed during NET formation, as the supernatant of control-treated neutrophils did not show substantial MPO-DNA levels, despite having measurable amounts of DNA and MPO (Fig 1A and Table 1).

In the present study, ELISA measurements of MPO-DNA complexes repeatedly demonstrated specificity and a concentration-effect relationship for *in vitro* generated NETs, as shown previously [46]. When spiked into plasma at low dilution, detection of this calibrator material was essentially abolished and still severely reduced at higher (1:100) plasma dilution, pointing to interference of plasma matrix components with the ELISA procedure (Fig 6). DNA digestion abolished the observed signal for *in vitro* generated NETs (Fig 5A) but left the signal for plasma samples detectable, therefore confirming the specificity for *in vitro* but not *in vivo* generated MPO-DNA complexes when assayed in plasma. In contrast, MNase treatment of plasma samples led to elevated signals for MPO antibody and for isotype coated wells (Fig 5B and 5C). Interestingly, limited digestion of sample DNA has previously been suggested to potentiate the assay signal and to increase assay sensitivity [33], but an isotype control was not performed in this study. Thus, it remains a matter of speculation what leads to elevated ELISA signals for plasma samples after DNA digestion. One could assume that DNA degradation facilitates access or interaction of the interfering plasma factor(s) with ELISA components.

In the attempt to identify and eliminate the interfering plasma factor(s), we first confirmed with uncoated ELISA microwells that the blocking step was sufficient and did not allow for unspecific absorption of sample components and/or detection antibody to the solid phase (Fig 1B and 1C). Also, unspecific binding of MPO by the isotype control antibody was excluded

(Fig 3A). We then hypothesized an unspecific reaction of one or more plasma component(s) with the coating antibody being responsible for the interference (Fig 10).

We started several attempts to resolve the issue of interfering plasma component(s). Exchange of assay antibodies (to antibodies from different species or by reversing the antibody pair as done for NE-DNA complexes [25]) yielded high background signals and low dynamic range, which was not observed for *in vitro* generated NET samples in the initial (standard) ELISA protocol, or even signals for uncoated wells upon application of plasma samples (Fig 2). Neither the change to a different antibody manufacturer nor to other mouse IgG isotype classes had any impact on the circumstance that plasma samples consistently yielded comparable optical densities for MPO antibody or isotype coated wells of the ELISA microplate (Fig 4A).

Also, exchange of ancillary ELISA components to entirely match a previously published protocol [28] did not yield substantial assay improvement regarding plasma samples (Fig 3C and 3D). Of note, additionally tested LowCross-Buffers were described to reduce frequently observed interferences in immunoassays by inhibiting low and medium affinity interactions but still permitting high affinity bindings [35, 36]. Application of these buffers reduced ELISA

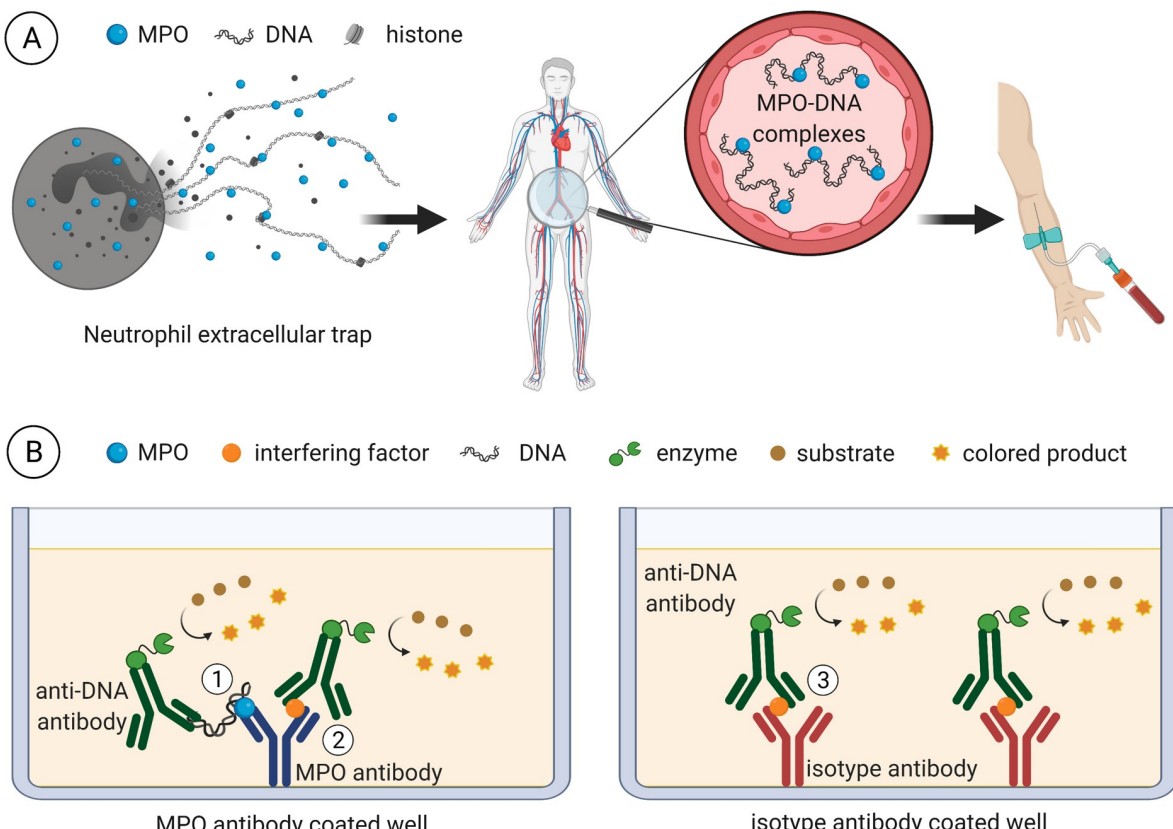

**Fig 10. Graphical illustration of MPO-DNA ELISA limitations in human plasma analysis.** A) During inflammatory conditions, NETs are formed locally. It is proposed that NET components such as MPO-DNA complexes are released into circulation and can be detected in plasma samples. B) When sandwich ELISAs are performed for MPO-DNA measurements, the capture antibody is commonly directed against MPO. The DNA moiety is then detected with an anti-DNA antibody conjugated to an enzyme for substrate turnover into a colored product (1). In plasma samples, a putative interfering factor (likely a multivalent substance) additionally mediates unspecific interactions between the two antibodies (2). This is revealed, when the capture antibody is exchanged for an isotype control and the interfering factor retains its capacity for immune complex assembly which is independent of the presence of MPO or DNA as documented by our analyses (3). The figure was prepared with BioRender online tool (Toronto, Canada).

signals to a minimum for both *in vitro* and *in vivo* generated NETs, possibly indicating that the MPO-DNA interaction is not strong enough to withstand these buffer formulations (S2C and S2D Fig). Alternatively, antibody binding may have been impaired by these buffers.

Immunoassay interference can be further provoked by multivalent substances such as heterophile antibodies. Multivalent antibody binding substances were determined in about 40% of serum samples and were calculated to cause, for example, falsely elevated human chorionic gonadotropin levels in 15% of the normal population if interference was not prevented [43]. Heterophile antibodies are natural antibodies or autoantibodies against weakly defined antigens, usually have low affinity and exhibit multi-species reactivity [37–39]. Similar influences can be exerted by high affinity HAAAs comprising the subgroup of HAMAs [37]. Immunoassays built with two mouse antibodies (as the present one) are particularly prone to exhibit HAMA effects [38, 39, 52]. Replacement of assay antibodies from mice with those from other species can reduce a HAMA effect [53], but the inclusion of a rabbit antiserum was technically not successful in our study (Fig 2A). We thus attempted the preadsorption of interfering substance(s) to isotype coated plates or removal of plasma immunoglobulins by protein A/G agarose which did not improve ELISA specificity for plasma contained MPO-DNA complexes (S3A–S3C Fig). It is controversially discussed whether usage of assay antibodies lacking Fc fragments reduces the risk of interference [54–56]. In our setting, however, this did not substantially ameliorate assay specificity for plasma samples (Fig 4E). Application of commercially available blocking reagents may help to prevent anti-animal and heterophile interference [38, 41, 52, 57, 58]. The addition of mouse IgG did not significantly improve the selectivity for MPO-DNA complexes in plasma (Fig 8B). Supplementation of sample diluent with TRU Block Ready enlarged the detection window between signals of MPO antibody and isotype coated wells (Fig 8D and 8E) when combined with high (1:100) plasma dilution. As this protocol modification was the only measure that improved the ELISA specificity for MPO-DNA complex detection in plasma samples, it was finally applied for investigation of human blood samples of acute and chronic inflammatory conditions despite the circumstance that intra- and interassay CVs were as high as 16% and 29%, respectively.

A human experimental endotoxemia model allowed us to compare baseline conditions in healthy volunteers with acute changes induced by low-dose LPS. As previously reported by others [59] we observed that the NET parameter citH3 rises significantly to 5-fold plasma levels peaking at 4 h after LPS challenge (Fig 9B). While cell-free DNA-histone complexes and neutrophil degranulation markers MPO and NE correlated significantly with plasma citH3 (thus documenting *in vivo* neutrophil activation and NET induction), MPO-DNA complex levels as determined by the initial or modified ELISA protocol did not increase during the time course (Fig 9C) nor correlate positively with any of the other markers (Table 2). A similar result was obtained with a second set of plasma samples from AAA patients (Table 3). Despite the fact that we have recently reported the circulating NET parameter citH3 to be significantly elevated in the chronic inflammatory setting of abdominal aortic aneurysms [44], plasma MPO-DNA complex levels determined by ELISA did not correlate with citH3. Furthermore, when we set the median MPO-DNA ELISA signal (determined with the modified protocol) of AAA patients (296 RU) and healthy donors of the experimental endotoxemia study at baseline (555 RU) in relation to the calibrator sample generated from isolated human neutrophils (1600 RU for $2 \times 10^6$ neutrophils/ml), the recorded values equaled $0.4–0.7 \times 10^6$ neutrophils/ml which would correspond to about 10% of a normal neutrophil blood count ($2–7.5 \times 10^6$/ml) stimulated for NETosis. Considering that a local rather than systemic NET induction at a much lower level would be expected, the recorded MPO-DNA values seem highly unlikely.

## Conclusion

Our efforts to improve the MPO-DNA ELISA protocol for plasma samples led to a detectable decrease in unspecific signals for isotype control which we further deducted from corresponding plasma MPO-DNA measurements to best reflect specific signals. When applied to a test set of plasma samples from experimental human endotoxemia or from AAA patients, ELISA signals were strong despite high plasma dilution. The essential lack of correlation between measured MPO-DNA levels and other neutrophil activation or NET parameters in this set of plasma samples leads us to question the suitability of the ELISA to reliably quantitate circulating MPO-DNA complexes reflecting *in vivo* generated NETs. In light of these findings, we propose that MPO-DNA assessment in plasma samples should be interpreted with caution, should preferably include an isotype control and be conducted in comparison to additional markers of NET formation. It remains to be elucidated whether ELISA analysis of DNA complexes with other neutrophil proteins such as NE [12, 25, 26] faces similar specificity issues for blood specimens. Of note, the comparison with a recently published ELISA protocol for citH3-DNA complexes [60] may give an explanation for the assay limitations: When we performed the recommended ELISA procedure with MPO versus citH3 capture antibodies, we found that the plasma level (ELISA signal) of histone-DNA complexes was substantially higher than for MPO-DNA complexes, thus providing a better window of detection between specific signals and unspecific reactions with isotype control which were also observed in the citH3-DNA assay. Thus, the quantity and likely also the quality (stability) of circulating DNA complexes with neutrophil proteins seems to restrict their assessment by ELISA and sufficiently high blood levels of MPO-DNA complexes might only be achieved during exceptional disease states.

## Supporting information

**S1 Fig. Variation of anti-MPO coating antibody concentration and biotinylated MPO detection reagents.** A) Microplate wells were coated with 5 μg/ml or 2.5 μg/ml MPO antibody. Supernatant of PMA-treated neutrophils was diluted (two-fold dilution series) and applied for assessment of MPO-DNA complexes using the initial ELISA protocol. Preparation of calibrator batch #1 was used in this experiment. B) Microwells were left uncoated, were blocked and then incubated with 1:2 diluted plasma samples. For signal development, a complete detection system consisting of 0.5 μg/ml biotinylated mouse anti-human MPO monoclonal antibody (no. HM2164BT, HyCult Biotech), 0.5 μg/ml streptavidin-HRP and ABTS substrate solution was sequentially applied. Where indicated, the biotinylated MPO antibody or additionally the streptavidin-HRP conjugate were omitted.
(TIF)

**S2 Fig. Exchange of ELISA ancillary components does not alter unspecific signals of plasma samples.** A) and B) Microwells were coated with MPO antibody, isotype control or were left uncoated and incubated with A) calibrator or B) 1:4 diluted plasma samples. The assay was performed according to the initial MPO-DNA complex ELISA protocol, but blocking buffer, sample diluent and detection antibody diluent were replaced by an in-house blocking buffer. C) and D) Microwells were coated with MPO antibody or isotype control. C) Calibrator or D) 1:5 diluted plasma samples were assayed using different sample and detection antibody diluents: IEP, diluents as outlined for the initial ELISA protocol; LCBM, LowCross-Buffer Mild; LCB, LowCross-Buffer; LCBS, LowCross-Buffer Strong. E and F) Microwells were coated with 5 μg/ml MPO antibody, isotype control or were left uncoated and incubated with E) calibrator or F) 1:5 diluted plasma samples using different reagents: for samples labeled "IEP", blocking buffer as well as sample diluent and detection antibody diluent were applied as

outlined for the initial MPO-DNA complex ELISA protocol whereas for samples labeled "The Blocking Solution", The Blocking Solution (Candor Bioscience) was used for blocking, sample dilution and detection antibody dilution. Note that data points for isotype control in E) are not visible since they presented with almost identical optical densities as for uncoated wells. (TIF)

**S3 Fig. Approaches to block multivalent interfering substances or HAMAs in plasma to improve the ELISA selectivity for MPO-DNA complexes.** A) Calibrator (at a relative concentration of 32) and plasma samples (at 1:2 dilution) were pre-adsorbed (+) or not (-) on isotype control coated and blocked microwells for 2 h. Samples were then transferred onto microwells coated with MPO antibody, isotype control or left uncoated and processed as indicated for the initial MPO-DNA complex ELISA protocol. B) and C) Plasma immunoglobulins were removed with protein A/G PLUS-agarose. Precleared plasma (+), samples comparably incubated without protein A/G (-) or fresh plasma (F) were then subjected to B) Western blotting or C) the initial MPO-DNA complex ELISA protocol using MPO antibody, isotype control or uncoated microwells. For ELISA, a final plasma dilution of 1:4 was applied. Expected molecular weights are: IgG heavy chains: 50 kDa (IgG1, IgG2 and IgG4) and 60 kDa (IgG3); IgM heavy chains: 70 kDa; IgG and IgM light chains: 23 kDa. M, molecular weight marker (Sharp Pre-stained Protein Standard, given in kDa). (TIF)

**S4 Fig. Comparison of different calculation methods for plasma MPO-DNA complexes in the experimental endotoxemia model.** Plasma samples of seven healthy volunteers at baseline (0 h) and at various time points after infusion of 2 ng/kg LPS were assessed for MPO-DNA complexes according to the initial (IEP) as well as the final, modified ELISA protocol (FEP). A) MPO-DNA concentrations were determined as outlined in materials and methods, i.e. the optical density obtained for the isotype control was first subtracted from the optical density of MPO antibody-coated wells. The resulting value was then used to calculate the MPO-DNA complex content according to the calibration curve. B) Additionally, an alternative calculation method was applied, since calibrator curves were non-linear with the inherent problem that net signal excerpted from a low relative concentration segment in the calibration curve would correspond to a greater concentration difference (between actual sample and its isotype control) than the same net signal excerpted from a high relative concentration segment on the calibration curve. Thus, the optical densities obtained for the isotype control and MPO antibody-coated wells were separately used to calculate concentrations in relation to the calibration curve. Thereafter, the established value for the isotype control was subtracted from the calculated concentration of MPO-antibody coated wells. C) For comparison, calculated MPO-DNA complex values without isotype control subtraction are provided. Medians and their 50% confidence intervals are indicated. ★ $p < 0.05$ according to Wilcoxon signed-rank test (in comparison to baseline); RU: relative units. (TIF)

**S1 Table. Concomitant analysis of control- and PMA-treated neutrophil supernatant by MPO-DNA ELISA.** (DOCX)

**S2 Table. AAA patient demographics: Categorical variables.** (DOCX)

**S3 Table. AAA patient demographics: Metric variables.** (DOCX)

**S1 Raw images.**
(PDF)

**S1 Data.**
(XLSX)

## Author Contributions

**Conceptualization:** Hubert Hayden, Christine Brostjan.

**Data curation:** Hubert Hayden.

**Formal analysis:** Hubert Hayden, Nahla Ibrahim, Christine Brostjan.

**Funding acquisition:** Christine Brostjan.

**Investigation:** Hubert Hayden, Nahla Ibrahim, Johannes Klopf, Branislav Zagrapan, Christian Schoergenhofer.

**Methodology:** Hubert Hayden, Lisa-Marie Mauracher, Lena Hell, Thomas M. Hofbauer, Anna S. Ondracek, Bernd Jilma, Irene M. Lang, Ingrid Pabinger.

**Project administration:** Hubert Hayden, Christine Brostjan.

**Resources:** Johannes Klopf, Wolf Eilenberg, Christoph Neumayer, Christine Brostjan.

**Supervision:** Christoph Neumayer, Christine Brostjan.

**Validation:** Hubert Hayden.

**Visualization:** Hubert Hayden, Johannes Klopf, Christine Brostjan.

**Writing – original draft:** Hubert Hayden.

**Writing – review & editing:** Hubert Hayden, Nahla Ibrahim, Johannes Klopf, Branislav Zagrapan, Lisa-Marie Mauracher, Lena Hell, Thomas M. Hofbauer, Anna S. Ondracek, Christian Schoergenhofer, Bernd Jilma, Irene M. Lang, Ingrid Pabinger, Wolf Eilenberg, Christoph Neumayer, Christine Brostjan.

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
