## [Decision Letter · Decision Letter 0]

19 Jan 2021

PONE-D-20-40740

ELISA detection of MPO-DNA complexes in human plasma is error-prone and yields limited information on neutrophil extracellular traps formed in vivo

PLOS ONE

Dear Dr. Brostjan,

Thank you for submitting your manuscript to PLOS ONE. After careful consideration, we feel that it has merit but does not fully meet PLOS ONE’s publication criteria as it currently stands. Therefore, we invite you to submit a revised version of the manuscript that addresses the points raised during the review process.

We look forward to receiving your revised manuscript.

Kind regards,

Yi Cao

Academic Editor

PLOS ONE

Journal Requirements:

2.PLOS ONE now requires that authors provide the original uncropped and unadjusted images underlying all blot or gel results reported in a submission’s figures or Supporting Information files. This policy and the journal’s other requirements for blot/gel reporting and figure preparation are described in detail at https://journals.plos.org/plosone/s/figures#loc-blot-and-gel-reporting-requirements and https://journals.plos.org/plosone/s/figures#loc-preparing-figures-from-image-files. When you submit your revised manuscript, please ensure that your figures adhere fully to these guidelines and provide the original underlying images for all blot or gel data reported in your submission. See the following link for instructions on providing the original image data: https://journals.plos.org/plosone/s/figures#loc-original-images-for-blots-and-gels.

Reviewers' comments:

Reviewer's Responses to Questions

**Comments to the Author**

1. Is the manuscript technically sound, and do the data support the conclusions?

Reviewer #1: Yes

Reviewer #2: Yes

Reviewer #3: Yes

Reviewer #4: Partly

Reviewer #5: Yes

2. Has the statistical analysis been performed appropriately and rigorously? 

Reviewer #1: Yes

Reviewer #2: Yes

Reviewer #3: N/A

Reviewer #4: Yes

Reviewer #5: Yes

3. Have the authors made all data underlying the findings in their manuscript fully available?

Reviewer #1: Yes

Reviewer #2: Yes

Reviewer #3: Yes

Reviewer #4: Yes

Reviewer #5: Yes

4. Is the manuscript presented in an intelligible fashion and written in standard English?

Reviewer #1: Yes

Reviewer #2: Yes

Reviewer #3: Yes

Reviewer #4: Yes

Reviewer #5: Yes

5. Review Comments to the Author

Reviewer #1: This paper reports a very comprehensive test of a standard ELISA that has been used to measure DNA-MPO complexes in patient plasma. The assays have been used in a number of published studies including some in high profile journals. The authors conclude “we would thus like to alert the NET research community that ELISA-based quantitation of MPO-DNA complexes in plasma samples may be error-prone.” I would suggest that this is too mild a warning. The series of excellent experiments (briefly summarized below) actually shows that the ELISA is dominated by non-specific reactions, and is therefore completely useless for clinical or biological studies. The study also has warning for any sandwich ELISA used to measure antigens in plasma.

I enthusiastically recommend publication, and I have only minor suggestions for improvement. I would suggest perhaps more strongly condemning the assay and the published results that have used it. Also I would suggest adding at the end of the abstract the more general warning about ELISA for plasma proteins.

In Fig. 1C the high signal with the isotype Ab control is clearly noted. This suggests that the signalfrom plasma is completely non-specific. However it is also striking that donors 1 and 2 are 10 X different: is this real and what is its significance?

Fig. 2 shows that different combinations of capture and detection Abs have poor detection of pure calibrator, except when MPO is detected with a two-step streptavidin. However this completely fails in diluted plasma.

Fig. 3 explores and eliminates some possible sources of non-specificity. Preadsorption did not help.

Fig. 4 shows that the non-specificity in plasma is not due to the Fc fragment, since Fab gave the same results.

Fig. 5 shows that DNAse eliminated the signal from the pure calibrator, as expected for detection by anti-DNAse, but left the large non-specific signal in plasma.

Fig. 6 is a very important experiment, where the ELISA is tested by spiking calibrator into plasma. The signal from the spiked calibrator was completely swamped in 1:4 plasma, and showed severe non-specificity even in 1:100 plasma.

Fig. 7 further explores plasma dilution, with no help.

Fig. 8 tests whether adding mouse IgG or TruBlock can improve specificity. They discovered one extreme condition where the non-specific signal from isotype Ab was small: 10% TruBlock added to 1:100 diluted plasma. Here I think it would be important to specify the concentration of calibrator, and discuss whether it is even close to the concentrations expected in 1:100 diluted plasma. Apparently not, since the assay showed no significant correlation with related markers in a large set of patient plasmas.

Reviewer #2: General Comments: This is an extremely well-designed and described analysis of the methodological limitations in quantifying the concentration of myeloperoxidase (MPO)-DNA complexes via ELISA in human plasma. Analysis of MPO-DNA in circulating blood is increasingly used as a systemic biomarker for the release of so-called neutrophil extracellular traps (NETs) from neutrophils which either accumulate at discrete vascular sites or which are activated during various coagulopathies. NETs are complexes of DNA, histones, and various neutrophil intragranule proteins (such as MPO) released via a highly regulated mode of lytic cell death. Physiological NET release has a host-protective role in the sequestration and killing of different microbial pathogens, but sterile NET release can be pathogenic. These investigators have previously described NET release as a likely pathogenic component of abdominal aortic aneurysms and thus seek to systemically evaluate the known or proposed circulating indicators of in vivo NET release. As indicated in the title and abstract, their careful analysis indicates that current ELISA-based protocols for quantifying MPO-DNA complexes in human plasma samples suffer from multiple complications and are thus an unreliable approach for evaluating NET release as a biomarker for different human vasculopathies.

Specific Comments:

1. Given its narrow and technical focus, the "readability" of the MS by non-experts would be improved by inclusion of a graphical "abstract" that illustrates: 1) the biology of MPO-DNA complex release during NETosis; 2) the setup of the routinely used sandwich ELISA for MPO-DNA; and 3) the likely actions of plasma components on limiting efficacy of the ELISA.

Reviewer #3: Summary: The manuscript by Hayden et al reports on the lack of specificity of a published ELISA for quantification of myeloperoxidase (MPO)-DNA complexes in human plasma. Much or even most of the data presented is focused on determining the source of the lack of specificity and/or figuring out a way achieve specificity in plasma samples. Ultimately the authors find a way to achieve modest assay specificity using a particular blocking buffer and then apply their modified assay to quantify MPO-DNA complexes in plasma samples from abdominal aortic aneurysm (AAA) patients, correlating these measurements with other known markers of neutrophil activation or neutrophil extracellular trap (NET) formation but ultimately finding a lack of correlation between MPO-DNA complexes and these markers.

Broad Scope Comments: Unfortunately it is not highly uncommon for bioanalytical methods developed specifically for unique applications in biomedical research to be poorly or improperly developed then reported in the literature after relatively little scrutiny and then, in the hands of others, provide uninterpretable, irreproducible or, at worst, misleading results. So it is not terribly shocking that by merely applying proper negative controls (antibody isotype controls) the authors discovered a major flaw in an ELISA that has been used by others in published biomedical research. The authors are to be congratulated for their careful use of this ELISA that resulted in this discovery.

This paper is unconventional in that much of it is devoted to methodological troubleshooting rather than unfolding a “story” of biomedical discovery. Nevertheless, because the ELISA being reported on was previously published (and in rather high profile journals as the authors point out), it makes for a valuable case study of how to contend with the lack of specificity in an antibody-based assay. Moreover, the workaround developed by the authors to achieve a modest degree of specificity in the assay combined with application of the assay to AAA patient plasma samples with the finding that MPO-DNA complexes are not correlated with other known markers of neutrophil activation or NET formation make it a story worth reporting in the scientific literature.

Prior to publication, however, numerous issues need to be addressed by the authors. Since one of these involves re-analysis of the MPO-DNA complex ELISA data from the AAA patients, a “major” revision was recommended. But barring any unforeseen problems, this reviewer feels that once the specific issues below are addressed, the manuscript should be published.

Specific Comments:

1) Methods; Fab fragment preparation: Additional technical details are needed regarding this separation so that readers can repeat it themselves if desired. For example, with the description given, it’s impossible to determine how, exactly, fractions were collected for the data in Fig. 4.

2) Table 1: The number of significant figures reported for many entries is too high. Most likely only 2-3 significant figure are warranted.

3) The data in Table 1 are derived from three different batches of neutrophils from two different donors. Please clarify which donor(s) contributed to which batches.

4) Lines 436-438 states, “Similar results were found for application of a commercially available blocker (The Blocking Solution, S2 Fig E-F).” The results in panel F are not at all similar to the results in panel D. In fact, these results seem to show in vivo specificity that is at least as good as the final solution worked out and presented in Fig. 8D and in Fig. S3. Why was this apparent finding seemingly disregarded? In a related matter, lines 695-697 state, “As this protocol modification [seen in Fig. 8D and Fig. S3D] was the only measure that improved the ELISA specificity for MPO-DNA complex detection in plasma samples, it was finally applied for investigation of clinical AAA samples.” Please also adjust this statement as needed in light of this critique.

5) Fig. 4: Indicate the band corresponding to antibody light chain in lane 2.

6) Line 483: There are no uncoated wells in panel B of Fig. 5. The statement in line 483 should refer only to panel C.

7) Line 498: Please refer to concentrations in the same terms in which they are displayed in Fig. 6. (Dilution factors are mentioned in the text but “relative concentration” is used in the figure.

8) For the final modified assay with moderate specificity that was developed, nonspecific signals are subtracted from specific signals providing a net signal that is used for relative quantification and comparison to other samples. (This is possible because of the isotype control that is run for each sample.) The problem with this approach is that the calibrator curves are non-linear. This means that the same net signal excerpted from a low relative concentration segment in the calibration curve will correspond to a greater concentration difference (between actual sample and its isotype control) than the same net signal excerpted from a high relative concentration segment on the calibration curve. The magnitude of this discrepancy may have influenced the conclusions drawn on the AAA patient samples. The data that went in to Table 2 should be re-processed and re-analyzed after taking this issue into account.

9) Lines 578-587: How many intra-assay sets were analyzed for both calibrators and controls? Also, please explain in greater detail exactly how multiple inter-assay variability experiments were conducted? Typically just one value is reported for this parameter. Also, explain in greater detail how control plasma samples were included on each single plate to adjust for high inter-assay variation. How did this adjustment work, exactly? Inter-assay variation on the order of 30% is quite high and is often considered unacceptable for certain types of measurements. Readers should be cautioned about this issue in the Discussion.

10) Line 614: For the high impact publications cited here, point out which ones did not use isotype controls.

11) This reviewer thinks that Fig. S3 (or at least panels D-G within it) should be part of the main text and not in supplemental information.

12) Line 966: Describing the data in Fig. S2 panel F as “1:5 diluted plasma samples using various buffers for blocking” is too vague. What is meant by "using various buffers for blocking"? There appears to be only one blocking buffer (or buffer combination) used in this experiment. Also, what was the antibody dilution in this experiment?

13) Please adjust the caption to Fig. 3 to better explain exactly where the isotype control data points and lines are (i.e., exactly where they are hidden in the figure).

14) Fig. 7: Please explain why the same dilution of the same donor plasma doesn't give the same (or similar) OD in the different graphs here. Is this simply due to the poor inter-assay precision?

Reviewer #4: The authors improved a published MPO-DNA ELISA method for plasma samples, since it showed low specificity for in vivo plasma samples. I think the work is interesting and meaningful. However, some issues should be presented clearer.

How author confirm the existed method was inaccurate? Is there any testing method for MPO-DNA to verified your opinion?

In result section, I think the process of improvement could be concentrated. And more optimization of new method and comparison between new and published methods could be discussed.

A conclusion section is needed.

Reviewer #5: In this manuscript, the authors present data on the assay on NETs using a coupled immunoassay. While this assay performed well with in vitro generated NETs, a number of technical issues were encountered using blood samples.

The authors explore in detail possible explanations for the technical issues and provide an approach to improve the assay.

The results are perhaps surprising in view of the many publications on the subject, raising questions as to why the problems were not previously detected.

One concern relates to the number of donors studied as well as the study of patients with abdominal aortic aneurysms. While patient with AAA may have NETs, it is hard to know how these values would compare with patients with infectious or autoimmune disease. I think therefore that it is important to study in the original as well as revised assay a larger panel of both control samples as well as samples with a broader range of autoimmune/inflammatory/infectious diseases. Since the literature on the latter conditions is likely larger than that on AAA, the value of the paper would be increased by allowing comparison with the current literature.

Given the implications of the study for the assay of NETs, additional analysis of samples is important.

6. PLOS authors have the option to publish the peer review history of their article (what does this mean?). If published, this will include your full peer review and any attached files.

Reviewer #1: No

Reviewer #2: No

Reviewer #3: No

Reviewer #4: No

Reviewer #5: No

---

## [Author Response · Author response to Decision Letter 0]

26 Mar 2021

Please refer to the submitted file "PONE-D-20-40740 Response to Reviewers".

---

## [Decision Letter · Decision Letter 1]

5 Apr 2021

ELISA detection of MPO-DNA complexes in human plasma is error-prone and yields limited information on neutrophil extracellular traps formed in vivo

PONE-D-20-40740R1

Dear Dr. Brostjan,

We’re pleased to inform you that your manuscript has been judged scientifically suitable for publication and will be formally accepted for publication once it meets all outstanding technical requirements.

Kind regards,

Yi Cao

Academic Editor

PLOS ONE

Additional Editor Comments (optional):

Reviewers' comments:

Reviewer's Responses to Questions

**Comments to the Author**

1. If the authors have adequately addressed your comments raised in a previous round of review and you feel that this manuscript is now acceptable for publication, you may indicate that here to bypass the “Comments to the Author” section, enter your conflict of interest statement in the “Confidential to Editor” section, and submit your "Accept" recommendation.

Reviewer #1: All comments have been addressed

Reviewer #2: All comments have been addressed

2. Is the manuscript technically sound, and do the data support the conclusions?

Reviewer #1: Yes

Reviewer #2: Yes

3. Has the statistical analysis been performed appropriately and rigorously? 

Reviewer #1: Yes

Reviewer #2: Yes

4. Have the authors made all data underlying the findings in their manuscript fully available?

Reviewer #1: Yes

Reviewer #2: Yes

5. Is the manuscript presented in an intelligible fashion and written in standard English?

Reviewer #1: Yes

Reviewer #2: Yes

6. Review Comments to the Author

Reviewer #1: (No Response)

Reviewer #2: The authors have comprehensively addressed all concerns and recommendations noted by the 5 reviewers of the initially submitted MS.

7. PLOS authors have the option to publish the peer review history of their article (what does this mean?). If published, this will include your full peer review and any attached files.

Reviewer #1: **Yes: **Harold P. Erickson

Reviewer #2: No

---

## [Editor Report · Acceptance letter]

12 Apr 2021

PONE-D-20-40740R1 

ELISA detection of MPO-DNA complexes in human plasma is error-prone and yields limited information on neutrophil extracellular traps formed *in vivo*

Dear Dr. Brostjan:

I'm pleased to inform you that your manuscript has been deemed suitable for publication in PLOS ONE. Congratulations! Your manuscript is now with our production department. 

Kind regards, 

on behalf of

Dr. Yi Cao 

Academic Editor

PLOS ONE